# Temporally-Delineated Sources of Major Chemical Species in High Arctic Snow

Katrina M. Macdonald[1], Sangeeta Sharma[2], Desiree Toom[2], Alina Chivulescu[2], Andrew Platt[2], Mike Elsasser[2], Lin Huang[2], Richard Leaitch[2], Nathan Chellman[3], Joseph R. McConnell[3], Heiko Bozem[4], Daniel Kunkel[4], Ying Duan Lei[1], Cheol-Heon Jeong[1], Jonathan P. D. Abbatt[5], Greg J. Evans[1]

[1]Department of Chemical Engineering and Applied Chemistry, University of Toronto, M5S 3E5, Canada
[2]Climate Research Division, Environment and Climate Change Canada, Toronto, M3H 5T4, Canada
[3]Desert Research Institute, Reno, 89512, Unites States of America
[4]Institute for Atmospheric Physics, Johannes Gutenberg University Mainz, Becher Weg, 21 55128, Germany
[5]Department of Chemistry, University of Toronto, M5S 3H6, Canada

*Correspondence to*: Greg J. Evans (greg.evans@utoronto.ca)

**Abstract.** Long-range transport of aerosol from lower latitudes to the high Arctic may be a significant contributor to climate forcing in the Arctic. To identify the sources of key contaminants entering the Canadian high Arctic an intensive campaign of snow sampling was completed at Alert, Nunavut, from September 2014 to June 2015. Fresh snow samples collected every few days were analysed for black carbon, major ions, and metals, and this rich data provided an opportunity for a temporally-refined source apportionment of snow composition via Positive Matrix Factorization in conjunction with FLEXPART potential emission sensitivity analysis. Seven source factors were identified: sea salt, crustal metals, black carbon, carboxylic acids, nitrate, non-crustal metals, and sulphate. The sea salt and crustal factors showed good agreement with expected composition and primarily northern sources. High loadings of V and Se onto Factor 2, crustal metals, was consistent with expected elemental ratios, implying these metals were not primarily anthropogenic in origin. Factor 3, black carbon, was an acidic factor dominated by black carbon but with some sulphate contribution over the winter-haze season. The lack of $K^+$ associated with this factor, Eurasian source, and limited known forest fire events coincident with this factor's peak suggested a predominantly anthropogenic combustion source. Factor 4, carboxylic acids, was dominated by formate and acetate with a moderate correlation to available sunlight and an oceanic/North American source. A robust identification of this factor was not possible; however atmospheric photo-chemical reactions, ocean microlayer reaction, and biomass burning were explored as potential contributors. Factor 5, nitrate, was an acidic factor dominated by $NO_3^-$, with a likely Eurasian source and mid-winter peak. The isolation of $NO_3^-$ on a separate factor may reflect its complex atmospheric processing, though the associated source region suggests possibly anthropogenic precursors. Factor 6, non-crustal metals, showed heightened loadings of Sb, Pb, and As, and correlation with other metals traditionally associated with industrial activities. Similar to Factors 3 and 5, this factor appeared to be largely Eurasian in origin. Factor 7, sulphate, was dominated by $SO_4^{2-}$ and MSA with a fall peak and high acidity. Coincident volcanic activity and northern source regions may suggest a processed $SO_2$ source of this factor.

# 1 Introduction and Background

Observations of Arctic climate have shown pronounced changes over recent years, including a rapid rise in surface temperature and the loss of sea ice and snow cover, with adverse local and global consequences (AMAP, 2017; Hartmann et al., 2013). Such changes in Arctic climate have been tied to contaminants within the Arctic atmosphere and snow, especially light absorbing compounds such as black carbon (BC) which can warm the surface and atmosphere (Clarke and Noone, 1985; Hansen and Nazarenko, 2004; Bond et al., 2013; Jiao et al., 2014). Furthermore, studies have found the long-range transport of lower-latitude anthropogenic and natural emissions to be a significant and substantial contributor to the Arctic aerosol burden (Stohl, 2006; Law and Stohl, 2007; AMAP, 2015). Thus, understanding the sources of these pollutants is a critical step in the development of control and mitigation strategies to protect the vulnerable Arctic environment.

The lower troposphere of the Arctic is separated from the upper and southerly atmosphere by a transport barrier known as the "Arctic front" or "Arctic dome". This dome is formed by surfaces of constant potential temperature, which inhibit the transport of southerly air masses into the lower Arctic troposphere, instead forcing northward-travelling air masses to rise over the dome. The size and location of the Arctic front is a complex system driven by global atmospheric conditions, with significant variation by season. Over the summer, the Arctic front is typically northward of 70 °N; however, during the winter the Arctic front extends farther south, as far as 40 °N (Stohl, 2006; Law and Stohl, 2007; AMAP, 2015). The Arctic front is also zonally asymmetric, typically extending much farther south over Eurasia during the winter. Thus, the Arctic atmosphere is more vulnerable to transport from southerly sources in the winter than the summer, especially Eurasian sources. Particles entering the Arctic atmosphere in winter can be removed only by atmospheric transport or deposition in snow where they can be retained for an extended time; thus, Arctic snow is a potentially critical reservoir within the Arctic system (AMAP, 2006). Given the seasonal variability in Arctic aerosol inputs and outputs, a period of enhanced accumulation is typically experienced during the Arctic winter and early spring termed "Arctic Haze". The haze is primarily composed of sulphate ($SO_4^{2-}$) and organic particulate matter with varying levels of ammonium ($NH_4^+$), nitrate ($NO_3^-$), mineral dust, and BC (Mitchell, 1957; Shaw and Wendler, 1972; Rahn, Borys, and Shaw, 1977; Barrie, 1986; AMAP, 2006; Quinn et al., 2007; and the references therein).

Interest in Arctic aerosol increased after the first observations of Arctic Haze in the 1950's (Mitchell, 1957; AMAP, 2006), and intensive routine monitoring of the Arctic atmosphere dates back to the late 1970's, particularly monitoring of BC and $SO_4^{2-}$ (Barrie, Hoff, and Daggupaty, 1981). However, direct measurements of pollutants in Arctic snow have been less common, particularly sampling campaigns of fresh snow. The relative abundance of Arctic aerosol data has facilitated extensive research on particulate sources (e.g., Sirois and Barrie, 1999; Stohl et al., 2013; Nguyen et al., 2013; Yttri et al., 2014). Fewer studies have identified the sources of snow impurities, which represent the deposited and surface albedo-influencing portion of the aerosol, and often these studies are reliant on modelled snow concentrations (e.g., Skeie et al., 2011; Wang et al., 2011) rather than measurements (e.g., Hegg et al., 2009; Hegg et al., 2010). Also, variability has been seen across existing snow apportionment studies. For example, previous studies show significant disagreement in the apportionment of BC during the Arctic winter, ranging from approximately 10% to over 90% attributed to biomass burning (e.g., Wang et al.,

2011 and e.g., Hegg et al., 2009, respectively). To the best of the authors' knowledge, no quantitative source apportionment has previously been conducted using temporally-refined fresh Arctic snow samples. Given the critical consequences arising from the deposition of BC and other impurities in snow, source apportionment specifically of these deposited chemical species is an important step towards understanding the Arctic environment. In this context, this paper analyzes the sources of chemical

components in freshly-fallen snow samples collected over a complete fall-winter-spring at a high Arctic location (Alert, Nunavut) and analyzed for a broad suite of analytes, using a combination of Positive Matrix Factorization diagnostics and Lagrangian dispersion modelling.

## 2 Methodology

### 2.1 Snow Sample Collection and Analysis

Sample collection and analysis was completed as per Macdonald et al. (2017). Briefly, fresh snow samples were collected at Alert, Nunavut (82°30' N, 62°20' W), from September 14th, 2014 to June 1st, 2015 from two snow tables located in an open-air minimal traffic site, about 1 km SSW of the Alert base camp. Replicate samples were collected after each snowfall, weather permitting, to a total of 59 sets of samples ranging from 1 to 19 days between samples with an average of 4 days. The use of a snow table allowed the deposition area associated with each sample to be recorded and used along with sample volume in the

conversion of measured concentration to flux. High winds in January and February may have led to undercatch of snow on the snow tables, under-estimating the calculated flux; however, snow composition measurements on these dates is not believed to have been impacted. Further details on the sampling methodology are provided in the supplemental section S1.

Snow samples were analysed for BC via single particle soot photometry (SP2), major ions via ion chromatography (IC) and pH analyzer, and soluble and insoluble metals via inductively coupled plasma mass spectrometry (ICP-MS). A summary of

the analysis methodology is provided in supplemental section S1.2, with further detail provided in Macdonald et al. (2017). Stringent quality assurances were followed throughout snow collection and analysis. The uncertainty of each measurement was estimated based on analysis detection limits and reproducibility as follows (Reff et al., 2007; Norris et al., 2014):

$$u_{ij} = \sqrt{\left( EF_j \ x_{ij} \right)^2 + \left( \frac{1}{2} MDL_j \right)^2} \ , if \ x_{ij} \geq MDL_j \tag{1}$$

$$u_{ij} = \frac{5}{6} MDL_j \ , if \ x_{ij} < MDL_j$$

$u_{ij} = 4 \bar{X}_j \ , if \ x_{ij} \ is \ missing$

where $x_{ij}$ is the $i^{th}$ measured value of analyte j, $u_{ij}$ is the uncertainty associated with this measurement, $EF_j$ is the error fraction for this analyte, $\bar{X}_j$ is the median measurement for this analyte, and $MDL_j$ is the method detection limit for this analyte.

The error fraction of each analyte was calculated as double the standard error of replicate measurements for each analysis, with a minimum of 10% imposed (Macdonald et al., 2017 as per Hegg et al., 2010). The method detection limit of each analyte was

calculated as three standard deviations of analyzed blank samples. The uncertainty for any samples with known preparation

concerns was doubled (e.g., partial sample melt in transit or poor mass closure over preparation); however, less than 7% of samples were noted as having potential preparation concerns.

The signal-to-noise (S/N) of each analyte was also calculated to indicate the strength of each measurement. Given the enhanced uncertainty of below MDL and missing values, these data points were excluded from the analysis per the suggestion of Norris et al. (2014) for environmental data. A S/N over 2 was considered to be strong, while a S/N from 0.2 to 2 was considered weak (Paatero and Hopke, 2003).

$$S/_{N_j} = \frac{1}{n} \sum_{i=1}^{n} d_{ij} \quad ; \qquad (2)$$

$$d_{ij} = \frac{x_{ij} - u_{ij}}{u_{ij}} \quad, if\ x_{ij} > u_{ij}$$

$$d_{ij} = 0 \quad, if\ x_{ij} < u_{ij}$$

where S/$N_j$ is the signal-to-noise of analyte j, n is the total number of samples, and $d_{ij}$ is a measure of the difference between the measured value and its uncertainty for the i[th] measurement of this analyte (all other variables are defined as in Eq. 1).

To complement snow measurements, simultaneous meteorological and atmospheric aerosol measurements throughout the campaign were considered, as provided by Environment and Climate Change Canada (ECCC). Local meteorological conditions were monitored by the Alert ECCC stations (Climate IDs 2400306, 2400305, and 2400302) (retrieved Nov. 2015 from climate.weather.gc.ca). Atmospheric composition was monitored at the Alert base camp: BC via hourly SP2 and major ions via IC of 6 to 8-day high-volume filters of total suspended particles (Hi-Vol). All data are presented in an earlier publication (Macdonald et al., 2017).

## 2.4 Computational Analyses

Two approaches to source identification were used. Source type was explored via measurement apportionment to identify the source composition and seasonal contribution. Source location was explored via backward particle dispersion modelling.

### 2.4.1 Source Apportionment

Positive matrix factorization (PMF) is a numerical technique for describing speciated data as factors with associated compositional and temporal profiles. This study uses the most recent US EPA version, PMF5, which uses the multilinear engine ME2 to solve the following equation set (Norris et al., 2014):

$$X = G \cdot F + E \quad, \qquad x_{ij} = \sum_{p=1}^{q} g_{ip} \cdot f_{pj} + e_{ij} \quad ; \qquad (3)$$

$$Q = \sum_{i=1}^{n} \sum_{j=1}^{m} \left( \frac{e_{ij}}{u_{ij}} \right)^2 \quad ,$$

where X is the n by m matrix of measurements with associated uncertainties u, G is the calculated n by q matrix of factor contributions, F is the calculated q by m matrix of factor compositions, E is the n by m error matrix, with lower case variables representing the specific value therein for the i[th] sample of the j[th] analyte for the p[th] factor, and Q is the object function.

So, for any dataset with n measurements of m analytes a solution is found for the matrices G and F for a particular number of factors, p, which produces the minimum value of Q, an optimization parameter calculated as the summed residual error, e, weighted by the measurement uncertainty, u. An additional 10% uncertainty was applied to all measurements in the PMF analysis, beyond that uncertainty captured in Eq. 1, to account for extra modelling uncertainty and further reduce the impact of noise. Any missing measurements were replaced with the median measured value (Norris et al., 2014).To determine the optimal number of factors, p, trial runs ranging from 2 to 9 factors were completed using 100 distinct random seeds per run. Trials were compared in terms of relative Q-value, improvement of solution with each additional factor, solution reproducibility, solution fit, and solution interpretability. Only solutions which produced factor profiles which could be explained in a real-world setting were considered. Random error and rotational ambiguity of the selected solution was explored by rerunning with 500 seeds, analysis of G-space plots, and quantification via the bootstrap error model. Supplemental section S1.2 provides further details on the PMF analysis.

The number of measurements included, n, was limited to dates with sufficient snowfall to complete the majority of analyses. Given the limited number of snow samples measurements available, a subset of the analyzed chemical species was used for PMF analysis. Only analytes with over 60% of measurements above MDL and strong S/N were included in the analysis. Analytes of particular interest to this study with sufficient S/N but only 30-60% of measurements greater than MDL were included in some cases but defined as weak variables (i.e., user-defined uncertainty was tripled for these analytes). Analytes which duplicated others were also excluded from the PMF analysis, such as analytes measured by two methods (e.g., IC and ICP-MS overlapping analytes) and analytes which are expected to share a common source and show extremely strong correlations (e.g., crustal metals with no significant anthropogenic source). Duplicate and closely related analytes do not provide additional information to the apportionment study but artificially inflate the importance of these analytes and increase the ratio of analytes to measurements unnecessarily. The complete list of chemical species included in each analysis is provided with the results.

### 2.4.2 Transport Modelling

The Lagrangian particle dispersion model FLEXPART, described in detail by Stohl et al. (2005), has been shown to be an effective tool for the prediction of transport pathways into and within the Arctic (e.g., Stohl, 2006; Paris et al., 2009). This potential emission sensitivity analysis was completed to identify likely source locations, as a complement to the PMF descriptions of source type. Modelled tracers were initialized over Alert and tracked backwards in time over a ten-day period at a 3-hr time step, driven using operational analysis data from the European Centre for Medium-Range Weather Forecasts with a horizontal resolution of 0.25° in longitude and latitude and 137 vertical hybrid pressure levels. Tracers were initialized at four altitudes over Alert: 100 m, 500m, 1000 m, and 2000 m above sea level. A simulation was completed for every 5 days over the campaign. Simulation results provided the expected residence time of the tracers at the horizontal resolution of the meteorological input data and on 10 levels up to 10 km.

The potential FLEXPART source regions associated with each PMF factor were identified. The peak periods associated with each factor, selected as the top 90$^{th}$ percentile of the factor contribution time series, were used to weight the FLEXPART ten-day residence times over the campaign , as per Eq. 4:

$$t_p^{xy} = \frac{\sum_{i=1}^{n} g'_{ip} \, t_i^{xy}}{\sum_{i=1}^{n} g'_{ip}} \tag{4}$$

$$g'_{ip} = g_{ip} \ , if \ g_{ip} \geq g_p^{90} \ else, \ g'_{ip} = 0$$

where $t_p^{xy}$ is the residence time at location x,y for p$^{th}$ PMF factor, $t_i^{xy}$ is the i$^{th}$ residence time at location x,y, $g_{ip}'$ is the 90$^{th}$ percentile contributions of the p$^{th}$ factor at time i, $g_p^{90}$ is the 90$^{th}$ percentile of $g_p$, and all other variables are as per Eq. 3. Only trajectories within 500 m of ground level were considered, given that low-altitude air masses are much more likely to show the influence of ground-level sources; however, selection of this 500 m cut-off height was found to have a negligible impact as the identified potential source regions were similar if it was adjusted in a sensitivity analysis by$\pm$ 300 m, although source regions for a higher cut-off height were expanded over a larger area. The weighted sum was then plotted to depict regions which likely influenced each factor. It should be noted that this approach will highlight the Arctic as a potential source for all factors given that all air masses were initialized at Alert within 500 m of the surface. Thus, interpretation of these plots must consider that they highlight both possible source regions as well as regions the air mass entered on route to Alert. Furthermore, factors with similar peaks will produce similar plots; however it was found that no factor of the selected solution shared more than two dates with peaks above their respective 90$^{th}$ percentile.

## 3 Results and Discussion

### 3.1 Optimal PMF Solution

Positive matrix factorization (PMF) was completed on 49 measurements of 20 analytes in Arctic snow (as listed in Table 1). Analysis of snow measurements as flux per period (i.e., the total deposited mass per area per snowfall) was found to be the most readily interpreted as physically realistic factors. PMF analyses of the snow measurements as concentration and flux per day are presented in the supplemental S3. Based on the criteria outlined in section 2.4.1, a seven-factor solution was found to be optimal. The seven-factor solution produced one of the largest Q-value improvements with the addition of a factor, an acceptable relative Q-value, and good reproducibility. In particular, the seven-factor solution showed a marked improvement in fit and interpretability over solutions with fewer factors.  The seven-factor solution reproduced measurements with a Pearson's correlation coefficient above 0.8 for all strong analytes. Furthermore, a repeat run using 500 seeds showed the seven-factor solution to be consistent and stable. The supplemental section S2 provides additional details on solution selection and the evolution of factor profiles over the completed runs. A brief overview of the results of the four and six-factor solutions is provided in the supplemental, as these solutions also showed merit as realistic apportionments of the data, although with poorer predicted/measured fit and residual error. Potential rotated solutions were considered, but showed no improvement over the

unrotated base solution. The final solution statistics are summarized in the supplemental section S2.3. The input and model diagnostic parameters for each analyte included in this PMF analysis are provided in Table 1. Only the portions considered as insoluble for metals measured by ICP-MS were included in this analysis (Al, V, Cu, As, Se, Sb, and Pb). Residuals of all analytes were found to be normally distributed, based on PMF5's Kolmogorov-Smirnoff test, with the exception of $NO_3^-$ and

5    V, although both appear visually to be close to a normal distribution.

**Table 1: Overview of PMF seven-factor solution input and diagnostic properties.**

| Analyte | Input Properties | | | Diagnostic Properties | | |
|---|---|---|---|---|---|---|
| | MDL (ppb) | Missing Data | Data Below MDL | Predicted/ Measured Fit | Normalized Residual Mean | Normalized Residual Deviation |
| Strong Analytes | | | | | | |
| BC | 0.042 | 0% | 0% | 1.00 | 0.01 | 0.20 |
| ACE | 4.4 | 0% | 4% | 0.90 | 0.08 | 0.85 |
| FOR | 1.2 | 0% | 0% | 0.83 | 0.13 | 0.77 |
| $Cl^-$ | 18 | 0% | 0% | 0.96 | 0.03 | 0.43 |
| $NO_3^-$ | 5.0 | 0% | 4% | 0.99 | 0.01 | 0.22 |
| $SO_4^{2-}$ | 18 | 0% | 0% | 0.99 | 0.01 | 0.20 |
| $Na^+$ | 18 | 0% | 4% | 0.99 | 0.02 | 0.38 |
| $NH_4^+$ | 5.0 | 0% | 2% | 0.85 | 0.07 | 0.65 |
| $K^+$ | 5.0 | 0% | 12% | 0.77 | 0.25 | 1.11 |
| $Mg^{2+}$ | 18 | 0% | 22% | 0.95 | 0.03 | 0.54 |
| Al | 30 | 8% | 27% | 0.99 | 0.00 | 0.42 |
| V | 0.027 | 8% | 10% | 0.97 | 0.09 | 0.57 |
| As | 0.010 | 8% | 0% | 0.93 | 0.09 | 0.76 |
| Se | 0.084 | 8% | 16% | 0.99 | 0.01 | 0.43 |
| Sb | 0.013 | 8% | 0% | 0.87 | 0.17 | 0.95 |
| Pb | 0.16 | 8% | 8% | 0.97 | 0.05 | 0.67 |
| Weak Analytes | | | | | | |
| MSA | 1.9 | 0% | 73% | 0.70 | 0.11 | 0.53 |
| $Br^-$ | 5.0 | 0% | 53% | 0.46 | 0.09 | 0.49 |
| $C_2O_4^{2-}$ | 18 | 0% | 63% | 0.76 | 0.01 | 0.18 |
| Cu | 0.23 | 8% | 20% | 0.49 | 0.13 | 0.55 |

**Notes:** ACE = acetate; FOR = formate; MSA = methanesulphonate. Predicted/Measured fit presented is Pearson's correlation coefficient. Metals with a charge are those measured by IC, others are insoluble portions measured by ICP-MS.

### 3.2 Factor Discussion

10    The seven PMF factors are described by their composition (Figure 1), contribution over time (Figure 2) and potential areas of influence and/or source regions (Figure 3). Error estimates provided for the percent apportionment of each analyte are the 25th and 75th bootstrap analysis percentiles. The bootstrap analysis correctly mapped over 96% of sub-sampled data for each factor, with the exception of Factor 7 which was correctly mapped for 76% of the bootstrapped runs. Furthermore, sensitivity runs and additional analysis (as described in the supplemental) corroborated the presented results. Details on the solution sensitivity

15    and validation analysis are provided in the supplemental.

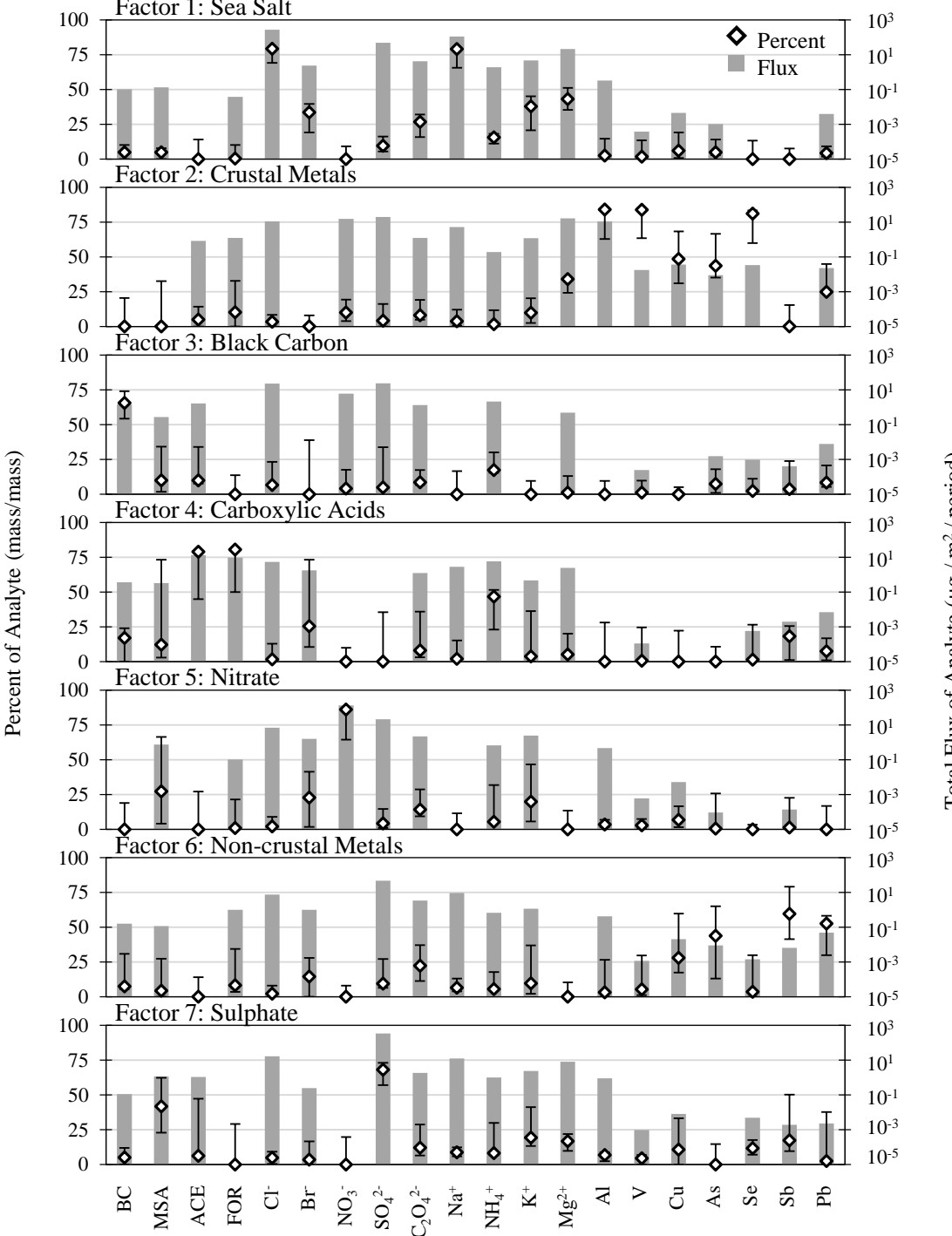

**Figure 1: Factor profiles. The loading of each analyte to each factor is provided as the portion of their flux apportioned to that factor as well as the percentage of the analyte's total flux (mass/mass) apportioned to that factor. Error bars on the percentage loading show the 25th and 75th percentiles of the bootstrapping analysis. Flux contributions below 0.00001 μg/m²/period are not shown. Metals with a charge are those measured by IC, others are insoluble portions measured by ICP-MS.**

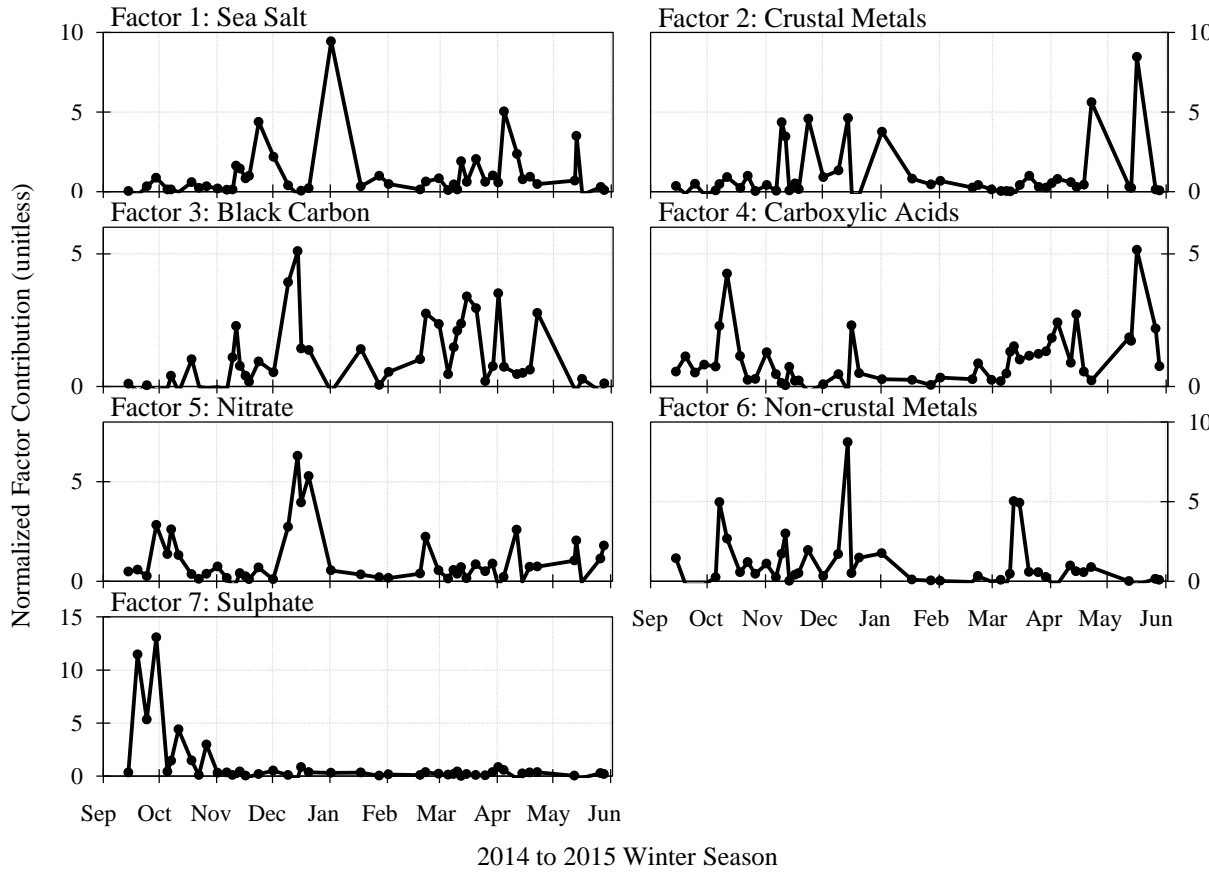

2014 to 2015 Winter Season

**Figure 2: Normalized factor contributions. The unitless contributions describe the relative magnitude of each factor over time such that the average contribution of each factor is one. For example, for the snowfall in early January Factor 1 sea salt had a contribution that was approximately ten times its average contribution over the campaign.**

The possible identities of each factor were suggested based on their composition, time series, correlations with non-apportioned analytes (i.e., analytes which were not included in the PMF apportionment) or with other measured parameters such as meteorology, and source regions. Table 2 summarizes the dominant analytes associated with each factor and their approximate breakdown of potential influencing regions. A neutralization ratio is also presented for each factor (i.e., $[Na^++NH_4^++K^++Mg^{2+}]$

10  / $[MSA+ACE+FOR+Cl^-+Br^-+NO_3^-+SO_4^{2-}+C_2O_4^{2-}]$, all as equivalence/m$^2$/period). Finally non-apportioned analytes and other measured parameters were correlated against the factor contribution time series and notable correlations are included in Table 2 below (Pearson's correlation greater than 0.7 are considered strong, and greater than 0.3 weak; listed in descending order).

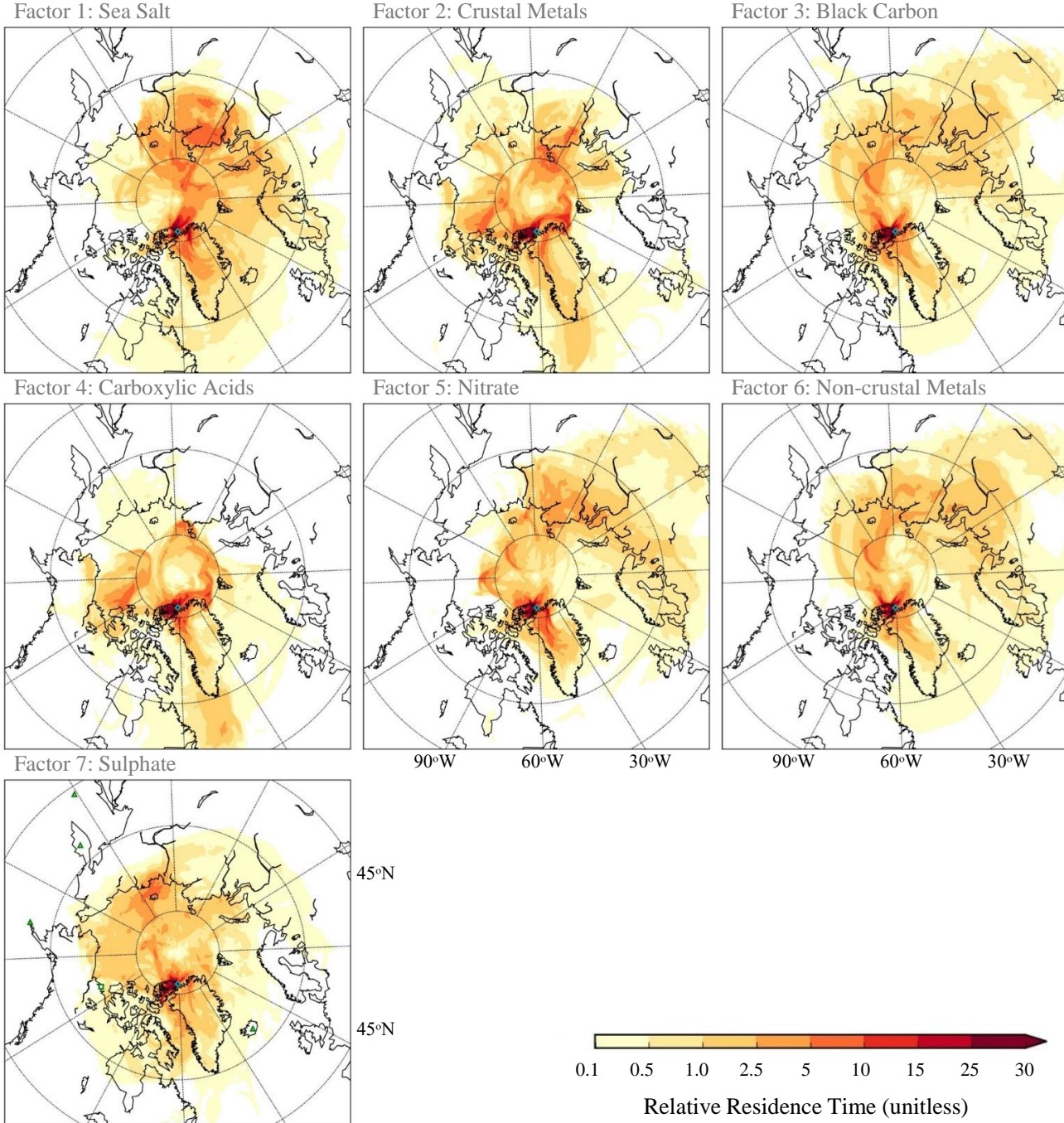

**Figure 3: Potential ten-day area of influence/source regions of apportionment factors. Cyan diamonds on plot shows the location of Alert, Nunavut. The Factor 7 sulphate plot depicts active volcanoes as green triangles, and the Smoking Hills as a green square (Hunter, 2007).**

**Table 2: Overview of factor characteristics.**

| Factor | Dominant Composition | Source/Influence Regions (% Residence Time) | | | | Neutralization Ratio | Peak Period(s) | Correlated Parameters |
|---|---|---|---|---|---|---|---|---|
| | | Arctic | North America | Eurasia | Open Ocean | | | |
| 1. Sea Salt | $Na^+$, $Cl^-$ | 84% | 1% | 14% | 2% | 0.79 [0.75-0.84] | episodic | Weak: period length |
| 2. Crustal Metals | Al, V, Se | 92% | 1% | 5% | 2% | 1.54 [1.17-2.11] | episodic | Strong: Fe, Mn, Co Weak: $Ca^{2+}$, Ti, basecamp winds |
| 3. Black Carbon | BC | 85% | 0% | 14% | 1% | 0.12 [0.17-0.38] | winter | Weak: Ti, As (soluble) Anti-corr.: temp. |
| 4. Carboxylic Acids | FOR, ACE | 94% | 2% | 1% | 3% | 1.02 [0.56-1.24] | fall/ spring | Weak: Ba, basecamp winds, propionate, sunlight |
| 5. Nitrate | $NO_3^-$ | 83% | 0% | 16% | 1% | 0.04 [0.03-0.19] | episodic | Weak: Ti, $H^+$, $NO_2^-$, |
| 6. Non-Crustal Metals | Sb, Pb, As | 82% | 0% | 17% | 1% | 0.36 [0.30-0.60] | episodic | Weak: Ti, As (soluble) |
| 7. Sulphate | $SO_4^{2-}$, MSA | 91% | 3% | 6% | 1% | 0.17 [0.15-0.30] | fall | Strong: $NO_2^-$ Weak: $H^+$, Ba, $Ca^{2+}$, temperature, sunlight |

**Notes:** **"Open Ocean" is defined as areas of the Atlantic and Pacific below 65 °N. "Arctic" source area includes the northern Pacific/Atlantic Oceans, Arctic Ocean, Canadian high Arctic, and Greenland. Neutralization ratio is described as: optimal solution [25th-75th bootstrapping]. Metals with a charge are those measured by IC, others are insoluble portions measured by ICP-MS unless noted as soluble.**

The factor characteristics and possible identifications are discussed in greater detail in the following sections.

### 3.2.1 Factor 1: Sea Salt

Factor 1 was found to resemble sea salt, primarily based on its composition. The first factor was characterized by high loadings (>75% of total flux mass apportioned to Factor 1) of $Na^+$ and $Cl^-$ and 30-45% loadings of $Br^-$, $K^+$, and $Mg^{2+}$ (Figure 1; Table 2). These dominant analytes and their relative proportions are consistent with that of sea salt (Pytkowicz and Kester, 1971), suggesting a marine origin for Factor 1. The ratios of $Cl^-$ and $K^+$ to $Na^+$ in Factor 1 were similar to that expected for sea salt with enrichment ratios close to unity of 1.3 and 1.1, respectively (1.3-1.4 and 0.8-1.2 25th-75th percentiles per bootstrapping analysis). The ratios of $Mg^{2+}$ and $SO_4^{2-}$ to $Na^+$ also resembled that of sea salt, with enrichment ratios of 1.6 and 1.7, respectively (1.5-1.7 and 1.0-1.9 25th-75th percentiles per bootstrapping analysis). Their slight elevation above unity may indicate some enrichment of these ions in the marine aerosol or inclusion of a separate source; however, a $Mg^{2+}$ enhancement of 1.6-1.7 was seen to be consistent among PMF analysis with greater/fewer factors. These enrichment ratios agree well with those measured by Krnavek et al. (2012): 1.33, 1, and 1.4 for $Cl^-$, $K^+$, and $Mg^{2+}$, respectively. The only sea salt analyte which was notably different from expected marine levels was $Br^-$ with an enrichment ratio of 3.4 (2.8-4.2 25th-75th per bootstrapping analysis), which may indicate aerosol enrichment of $Br^-$ relative to $Na^+$ or may be a result of this analyte's high uncertainty and poor

signal-to-noise. As shown in the supplemental Figure S1, apportioned Br⁻ was underestimated particularly in spring, suggesting that Br⁻ spring photo-chemistry (as per that observed by Toom-Sauntry and Barrie, 2002; Pratt et al., 2013) is not well-captured in this PMF analysis.

Factor 1 showed sporadic peaks throughout the campaign, with the largest peak early January (Figure 2). The January peak coincided with a local blizzard, which may indicate local marine sources such as open water, blowing saline snow, or frost flowers. A strong correlation between high winds and salt emissions from fresh sea ice frost flowers has been suggested by others (e.g., Xu et al., 2013). However, Factor 1 also showed a moderate correlation with collection period length (Pearson's correlation coefficient of 0.47) and the noted January peak was one of the longest collection periods in the campaign This may indicate continuous dry/wet deposition of sea salt to the snow table over time. Weighting the FLEXPART predicted source areas by the Factor 1 peak dates (Figure 3) showed the Eurasian coast of the Arctic Ocean, the Norwegian Sea, the Greenland Sea, and the northern Atlantic Ocean to be potential sources of sea salt to Alert. Ice-free areas were identified using the NOAA G02135 ice concentration images (retrieved from ftp://sidads.colorado.edu/DATASETS/NOAA/G02135/ November 2017). During periods of peak Factor 1 contribution the East Siberian Sea, Laptev Sea, and Kara Sea appeared to have been largely ice-covered; however, the Barents Sea, Greenland Sea, Norwegian Sea, northern Atlantic, and portions of Baffin Bay and waters surrounding the Queen Elizabeth Islands all seem to have been ice-free or with new, thin ice coverage. Thus, sea salt spray from these areas likely contributed to the sea salt signal at Alert.

The marine factor was found to be highly robust over this apportionment analysis. All runs with more than two factors exhibited a resolved Na/Cl-dominated factor, and the composition of this marine factor was found to be maintained across each addition of a new factor with Pearson's correlation coefficients above 0.98. Bootstrapping analysis found little error associated with this factor. Furthermore, similar marine factors have been observed in previous apportionment studies of Arctic snow (Hegg et al., 2009; Hegg et al., 2010) and Arctic aerosol (Sirois and Barrie, 1999; Nguyen et al., 2013).

### 3.2.2 Factor 2: Crustal Metals

Factor 2 was characterized by elevated levels of Al, V, and Se, all over 80% mass/mass (m/m) loading, and 25-50% loading of Cu, As, $Mg^{2+}$, and Pb (Figure 1; Table 2). These metals suggest a crustal origin for this factor. The composition of dust is far more variable than that of sea salt; thus, no single enrichment ratio can be determined for each analyte loaded on to Factor 2. However, the modelled ratios of Al to $Mg^{2+}$, $K^+$, V, Cu, As, Se, Sb, and Pb all appear realistic when compared with a variety of crustal sources, with calculated enrichment ratios in the range of 1 to 15 (Taylor, 1964; Barrie, den Hartog, and Bottenheim, 1989; Masson-Delmotte et al., 2013). Specifically, the modelled ratio of As/Al (0.00081 m/m) was seen to be closer to that of local soils (0.00013) (Barrie, den Hartog, and Bottenheim, 1989) than the global typical composition (0.00002) (Taylor, 1964; Masson-Delmotte et al., 2013) with enrichment ratios of 6 and 37, respectively (6.3-9.5 and 37-58 25th-75th percentiles per bootstrapping analysis). The composition of Factor 2, crustal metals, suggested an alkaline aerosol with a neutralization ratio of 1.5, calculated as described in section 3.2.

Factor 2 showed sporadic peaks over the campaign but primarily from November to February and after April. This time series showed good agreement with non-apportioned metals typically considered to be dominated by crustal origins: insoluble Fe, Mn, Co, Tl, and Ca. The time series of this factor also showed slight correlations with winds from the direction of the base camp and winds speeds, with Pearson's correlations of 0.39 and 0.26, respectively. This, along with the calculated As/Al ratio,

suggest that this crustal factor may be dominated by local soil and dust, likely from cleared or paved areas at the Alert base camp. The potential source regions calculated from FLEXPART results for this factor are shown in Figure 3. Arctic areas dominate the identified potential source region, again suggesting local soils were a major contributor to this factor; however, potential long-range sources of northern Asia, North America and Atlantic Ocean were also identified. A primarily local dust source is supported by the findings of Zwaaftink et al. (2016), which showed that surface dust loads within the high Arctic are

typically dominated by Arctic sources with an annual average contribution of 70% deposited mass from sources above 60 °N. A factor dominated by crustal metals was consistently resolved among the completed apportionment runs. This factor maintained a fairly similar composition across all numbers of factors, with Pearson's correlation coefficients above 0.97. However, metals traditionally considered to be associated with industrial activities, such as Pb, Cu, As, and Sb, were observed to gradually split from this factor with the addition of new factors. The seven-factor solution for this factor showed low levels

of error according to the bootstrapping analysis. Similar factors have been observed in previous atmospheric apportionment studies (e.g., Sirois and Barrie, 1999; Nguyen et al., 2013) but typically were not seen to account for such a large percentage of these metals, i.e., with loadings of 25-60% m/m for major crustal analytes. This might suggest that a separate source was missed by this study, though this seems unlikely given the consistency of the observed factor. Sirois and Barrie (1999) found the crustal signature at Alert to be dominated by local sources during the fall and long-range transport in the late spring to

summer. This supports a single local source for this study focussed on the winter season, while studies of the full year may have split crustal analytes among various long-range transport sources. However, the April-May peak in the crustal factor observed in this study coincides with Sirois and Barrie's (1999) peak considered to be dominated by long-range dust transport.

### 3.2.3 Factor 3: Black Carbon

The third factor was characterized by a high loading of BC, 66% m/m of the total BC, 17% m/m loading of $NH_4^+$, and all other

analyte loads below 10% (Figure 1). BC is a combustion product from both fossil fuel and biomass burning. While $NH_4^+$ is more commonly associated with agricultural emissions, it can also be produced by biomass burning, vehicle emissions, and some industrial activities (Behera et al., 2013). Most conspicuous in the composition of Factor 3 was the absence of $K^+$, considered to be a tracer of biomass burning which can be a significant source of BC. This separation of BC and $K^+$-rich factors was persistent for all PMF solutions with four or greater factors. Furthermore, the ratios of $SO_4^{2-}$ and $NO_3^-$ to BC were much

higher than would be expected for biomass burning with enrichment ratios above ten (Turn et al., 1997; Hays et al., 2005; Saarikoski et al., 2007; McMeeking et al., 2009; May et al., 2014); the relative loading of $SO_4^{2-}$ compared to $NH_4^+$ and $NO_3^-$ was also higher than expected for biomass burning (Liu et al., 2017). This factor also showed an acidic signature with a neutralization ratio of 0.12. Factor 3, BC, showed an enhanced contribution over the Arctic Haze season, November through

April (Figure 2). The time series of this factor did not show a strong correlation with any non-apportioned analyte; however, it did show weak correlations with insoluble Ti and V, and soluble As. The observed winter enhancement of this factor did not suggest a significant contribution from forest fires which are more prevalent in warmer months. Furthermore, the peaks in this factor did not coincide with dates of known northern hemisphere forest fire activity (as per fire records of NASA Global Fire

Maps, retrieved May 2016 from https://lance.modaps.eosdis.nasa.gov). Thus, the composition and time series of Factor 3, BC, suggested a predominantly anthropogenic combustion source with little contribution from biomass burning. A BC-dominated factor was resolved for all runs with four or more factors, and its composition remained consistent with Pearson's correlation coefficients of 0.95 or greater. Thus, the practically unique origin of BC was fairly robust through this analysis.

There have been many studies exploring the sources of Arctic BC, though primarily focussed on aerosol. The relative

importance of fossil fuel combustion and biomass burning differs between studies, likely indicating a strong dependence on location (especially high vs low Arctic), season, or differences in source breakdown year-to-year (e.g., McConnell et al., 2007; Doherty et al., 2010; Dou et al., 2012; Law et al., 2014). A recent study by Xu et al. (2017) analysing airborne measurement from a similar time period as this study found about 90% of BC to likely be anthropogenic in source, primarily from Eurasia, supporting the assessment above. Several modelling studies have suggested that combined anthropogenic sources account for

65-96% m/m of BC in Arctic snow, especially elevated over the winter months with spring and summer proportions dependent on the frequency of forest fires of that year (Flanner et al., 2007; Skeie et al., 2011; Wang et al., 2011; Sharma et al., 2013; Breider et al., 2014; Xu et al., 2017). In particular, modelling studies have shown winter Arctic BC to be dominated by flaring and other mixed industry emissions, with less impact from anthropogenic biomass burning (Flanner et al., 2007; Stohl et al., 2013). Studies of Arctic snow/aerosol composition have suggested that over 85% of BC is from the combustion of fossil fuels

year-round, based on radiocarbon analysis and measured ratios with biomass burning tracers (e.g., Slater et al., 2002; Yttri et al., 2011; Yttri et al., 2014; Barret et al., 2015). Hegg et al. (2009 and 2010) completed snow PMF apportionment analyses on spatially-defined samples. Unlike the majority of studies discussed above, these apportionment studies of Arctic spring snowpack attributed over 90% of BC to biomass burning origins. The Hegg et al. (2009 and 2010) studies showed the variability in BC sources to snow by location and season; however, in nearly all cases, including aged winter snow, pollution

sources were found to be small contributors relative to biomass burning. This may indicate a significant fluctuation in BC sources between years; however, both the 2014/15 season and the years of interest in the Hegg et al. studies (2009 and 2010) were found to represent fairly typical years in Northern hemisphere biomass burning emissions (Global Fire Emissions Database, version 4.1, retrieved July 2016 from http://www.globalfiredata.org). Thus, the findings of this study stress the variability in BC sources to Arctic snow and the importance of further measurements to better classify the main contributions.

Weighting of the FLEXPART results by the peak periods for Factor 3 indicated that central Eurasia was a probable source of this factor, especially northern Russia and some portions of central southern Russia (Figure 3). Russian industrial activities are known to be a significant source of BC; in particular, some studies have estimated about 70% of Russian BC emissions are related to flaring and transportation (Evans et al., 2017). These source regions correspond with known flaring and industrial BC sources (in the vicinity of the Ob and Pechora rivers and the Taymyrsky Dolgano-Nenetsky District, respectively) (Huang

et al., 2015; Winiger et al., 2017). A winter central Eurasian source of BC was also identified by Xu et al. (2017), specifically flaring in western Siberia. However, this central Asian source was found to be a smaller contributor of total Arctic BC than eastern Asia. Similarly, Hirdman et al. (2010) and Stohl et al. (2007) also identified a primarily northeastern Eurasian source to Arctic BC in the winter/late spring. The lack of a distinct eastern Asian source for Factor 3, BC, may indicate that the ten-day trajectory analysis was not long enough to fully capture this influence.

### 3.2.4 Factor 4: Carboxylic Acids

Factor 4 was characterized by high loadings, 79-80% m/m, of acetate (ACE) and formate (FOR); moderate loadings, 25-50%, of $NH_4^+$ and $Br^-$; and lower loadings, 10-20%, of Sb, BC, and MSA (Figure 1; Table 2). The loadings of $Br^-$ and MSA on this factor were highly variable, as shown by the bootstrap results in Figure 1; however, both of these analytes had high associated uncertainty. The composition of this factor suggested a neutral aerosol with a neutralization ratio of 1.02. Factor 4 exhibited peaks in October and May (Figure 2) and was seen to moderately correlate with propionate, hours of sunlight, and base camp winds, with Pearson's correlation coefficients of 0.4-0.5. Weighted FLEXPART results indicated local, North American, and Atlantic Ocean areas of potential influence for this factor (Figure 3). A variety of potential source may have contributed to this factor but the available evidence does not allow a robust identification. Possible contributors hypothesized in other studies of arctic carboxylic acids are discussed below including biomass burning, atmospheric or snow photochemical processing, and ocean microlayer emissions. However, some studies have postulated the existence of a yet unidentified source of high-latitude carboxylic acids (e.g., Paulot et al., 2011) which may be reflected in Factor 4. A similar high carboxylic acid factor was resolved for runs with six or greater factors and maintained its composition with Pearson's correlation coefficients of over 0.96; however, the loading of BC and $K^+$ onto this carboxylic acid factor was much more variable over the additional runs.

Carboxylic acids within the Arctic have previously been linked with biomass burning plumes (e.g., Jaffrezo et al., 1998; Legrand and de Angelis, 1996). The ratio of BC and $K^+$ apportioned to this factor was similar to that of a biomass burning plume, particularly the high $K^+$ proportion typical of herbaceous burning (Turn et al., 1997; Saarikoski et al., 2007; McMeeking et al., 2009; May et al., 2014). However, both BC and $K^+$ loading showed significant uncertainty. The loadings of formate, acetate, $Cl^-$, $Br^-$, $C_2O_4^{2-}$, and $NH_4^+$ appeared to be higher than expected for biomass burning emissions; the ratio of these analytes to BC were enriched by a factor of 3 to 75 relative to typical ratios of biomass burning emissions, based on a review of measured herbaceous and woody emissions (Turn et al., 1997; Andreae and Merlet, 2001; Hays et al., 2005; Saarikoski et al., 2007; McMeeking et al., 2009; May et al., 2014). The observed enrichment ratios of this factor above typical biomass burning plumes could be explained by atmospheric processing, for example, the cloud processing suggested by Legrand and de Angelis (1995). Alternatively, gas-phase partitioning and the subsequently enhanced scavenging observed in a previous study of this data (Macdonald et al., 2017) may have led to increased levels of some co-emitted chemical species relative to BC. The fall and spring peak of Factor 4 may support a biomass burning identification, as burning events are more typical in warmer seasons, specifically a North American source as suggested by the FLEXPART analysis.

Previous studies have also suggested a photochemical processing source of these carboxylic acids in the Arctic. Dibb and Arsenault (2002) found elevated levels of formic and acetic acid in the pore space of deposited Arctic snow and hypothesized oxidation of carbonyls and alkenes within the snowpack as a likely source. The prevalence of the factor in the fall and spring, before polar sunset and after polar sunrise, would support a photochemical source. Furthermore, summertime measurements of Arctic atmospheric samples by Mungall et al. (2017) also showed high levels of formic and acetic acid and hypothesized an oceanic microlayer photochemical source. Again, the temporal trend of Factor 4 as well as the Atlantic Ocean source location would support this possibility. An atmospheric budget analysis by Paulot et al. (2011) identified a significant missing source of high-latitude formic and acetic acid. Factor 4 of this study could represent a combination of the suggested sources above, or a missing source which is as yet unidentified.

### 3.2.5 Factor 5: Nitrate

The fifth factor was characterized by high $NO_3^-$, 86% m/m loading (Figure 1). This factor was also seen to have moderate loadings of MSA and $Br^-$, 20-30%, but with a larger degree of uncertainty. The atmospheric chemistry of $NO_3^-$ is complex, involving a variety of sources, formation mechanisms, and destruction mechanisms; in particular, snow can act as both a sink and a source of atmospheric nitrogen oxides, further complicating the local $NO_3^-$ cycle (Beine et al., 2002; Ianniello et al., 2002; Morin et al., 2008; Fibiger et al., 2016). Furthermore, the complex processing of $NO_3^-$ was demonstrated in the earlier deposition analysis of this data which suggested that gas-phase deposition was a dominant mechanism of $NO_3^-$ transport to snow (Macdonald et al., 2017). Thus, the loading of $NO_3^-$ onto a separate factor may be a reflection of its unique atmospheric processing. Comparison of simultaneous snow and atmospheric measurements over this campaign, as described in Macdonald et al. (2017), showed $NO_3^-$ to have a higher effective deposition velocity than BC or $SO_4^{2-}$. This supports external mixing of these compounds in the atmosphere and thus their assignment to separate source factors. This $NO_3^-$-loaded factor was resolved for simulations with six or greater factors, prior to which this factor appears to be combined with the carboxylic acid factor. In addition, a similar unique $NO_3^-$ factor was also observed in previous snow and atmospheric apportionment studies (Sirois and Barrie, 1999; Hegg et al., 2009; Hegg et al., 2010).

Factor 5 showed a variable contribution throughout the campaign but especially elevated in December. This factor was not found to correlate significantly with any non-apportioned analyte or meteorological parameter; however, Factor 5 did weakly correlate with nitrite ($NO_2^-$) and $H^+$ with Pearson's correlation coefficients of 0.35 and 0.46, respectively. This correlation with $H^+$ is in agreement with this factor's low neutralization ratio of 0.04. The low fall/spring levels of this factor may reflect the loss of $NO_3^-$ from snow through photolysis driven by the sunlight availability after polar sunrise (Morin et al., 2008; Fibiger et al., 2016). The highest $NO_3^-$ levels were observed when photolysis was inhibited during the polar sunset from mid-October to late-February. The movement of $NO_3^-$ accumulated in the snow to atmosphere during the spring is supported by the broad peak in atmospheric $NO_3^-$ observed via Hi-Vol filters from February to the end of the atmospheric sampling in mid-May. February to June, 2015, was also characterized by a "bromide explosion", observed as a broad peak in snow and atmospheric $Br^-$ (Macdonald et al., 2017). It is possible that this offered a different formation pathway for $NO_3^-$ over this period via the

reaction of $NO_2$ and BrO (Morin et al., 2008). The mid-winter peak in this factor may be linked to $NO_3^-$ formation via $N_2O_5$ hydrolysis in the aerosol phase, which is considered to dominate Arctic $NO_3^-$ chemistry during the night (Morin et al., 2008). Potential source areas of this factor, largely driven by the December peak, are shown in Figure 3. This plot was found to be similar to that of Factor 3, primarily northern Eurasia, though with a possible stronger dependence on northern Europe. Thus, 5 the $NO_3^-$ precursors to this factor may be largely anthropogenic in origin. Additionally, this factor appears to coincide with increased transport over the ice-free open water. This transport pathway might explain the presence of MSA, typically considered an indicator of marine biogenic activity within warmer ice-free water bodies (Li et al., 1993; Ye et al., 2015).

### 3.2.6 Factor 6: Non-Crustal Metals

Factor 6 showed a high loading of Sb, Pb, and As, 40-60% m/m, and moderate Cu loading, 28%, as shown in Figure 1. These 10 metals are frequently associated with industrial emissions, particularly high-temperature activities such as fossil fuel combustion and smelting (Berg, Røyset, and Steinnes, 1994; Laing et al., 2014). Although total Se and V loadings to this factor are low, the non-crustal loading of these metals (i.e., percentage of total excluding that which is loaded on Factor 2) are 20-30%. This factor also contains 10% of non-sea salt $SO_4^{2-}$. These constituents also point towards an industrial source (Berg, Røyset, and Steinnes, 1994; Laing et al., 2014). A neutralization ratio of 0.37 for this factor suggested an acidic aerosol. Factor 15 6 exhibited major peaks in October, December, and March and was found to be associated with a Eurasian source (Figure 2 and 3). Although several of the non-apportioned metals had limited measurements above MDL, a possible correlation was observed between Factor 6 and insoluble Ti, Cr, and Tl, and soluble As, Pb, Cr, and Cd. These metals are often considered to be primarily industrial in origin (Berg, Røyset, and Steinnes, 1994; Laing et al., 2014). The similarity in the FLEXPART potential source maps between Factor 3 BC and Factor 6 may support their mutual designation as anthropogenic-related.

20 Factors 6 and 7 were resolved separately only for solutions with seven or more factors. With the addition of a ninth factor, the non-crustal metals factor was further split into a factor dominated by As and Pb and a second factor dominated by Sb. This may represent the resolution of different industrial sources; however, the addition of these factors was not found to greatly improve the overall solution fit. Factors dominated by non-crustal metals, specifically Pb and As, have been observed in previous atmospheric apportionment studies (Sirois and Barrie, 1999; Nguyen et al., 2013) but not as clearly in existing snow 25 apportionment studies (Hegg et al., 2009; Hegg et al., 2010).

### 3.2.7 Factor 7: Sulphate

Factor 7 was characterized by $SO_4^{2-}$ and MSA, with loadings of 68% and 42% m/m, respectively (Figure 1). MSA is considered to be a tracer for biogenic marine activity; however, the ratio of $MSA/SO_4^{2-}$ observed in Factor 7, 0.003, is far below that typically seen for marine biogenic emissions, 0.05-0.20 (Li et al., 1993). The portion of $SO_4^{2-}$ in this factor related to marine 30 biogenic emissions was estimated at about 2-7%, assuming a typical MSA/marine-$SO_4^{2-}$ ratio and similar scavenging of MSA and marine $SO_4^{2-}$. Thus, an additional source of $SO_4^{2-}$ to this factor was suggested. Sulphate is typically an indicator of anthropogenic activities; however, the potential source regions identified for Factor 7 in Figure 3 are largely confined to the

Arctic where anthropogenic sources are minimal. The source region identified is only a ten-day back trajectory, so it is possible that the area of influence would extend farther south if longer trajectories were considered. However, given that 91% of the ten-day FLEXPART area is within 65 °N, a northern and likely natural source seems likely. Furthermore, BC and $NO_3^-$, typical indicators of industrial activity, both showed low loadings (<6% m/m) onto Factor 7, again suggesting this factor is not

anthropogenic in origin. The loading of $SO_4^{2-}$ onto Factor 3 BC and Factor 6 non-crustal metals are more consistent with anthropogenic sources.

Aside from anthropogenic and marine sources, volcanic activity can be a significant source of atmospheric $SO_4^{2-}$. Volcanic emissions are characterized by high levels of sulphur dioxide ($SO_2$, an oxidation precursor of $SO_4^{2-}$), acidic compounds, and a variety of metals (AMAP, 2006). A volcanic source would be consistent with the observed low levels of BC and $NO_3^-$

associated with Factor 7. Significant loadings of non-crustal Se and V (62% and 28% m/m, respectively), correlation with $H^+$ and Ba, and an acidic neutralization ratio also support a potential volcanic source for Factor 7 (Key and Hoggan, 1953; Rahn, 1971; Berg, Røyset, and Steinnes, 1994; AMAP, 2006; Laing et al., 2014). Several volcanoes within the near Arctic were known to be active over the 2014-15 season: Bárðarbunga, Iceland; Shishaldin, Aleutian Islands; Sheveluch, Bezymianny, and Zhupanovsky, Kamchatka Peninsula; and Chirpoi, Kuril Islands (Global Volcanism Program, retrieved March 2016 from

http://volcano.si.edu/). The Smoking Hills, naturally combusting coal and oil shale deposits on the northern coast of the Northwest Territories, Canada, at Cape Bathurst, 69.5 °N, 126.2 °W, are also located near the identified source region of Factor 7 (Freedman et al., 1990; AMAP, 2006); however, it is expected that the contribution from these hill would be minimal. Factor 7 showed a distinct maximum in September/October and a low contribution throughout the remainder of the campaign. This corresponds with volcanic activity at the Bárðarbunga volcano in Iceland as observed by others (Icelandic Met Office via

Global Volcanism Program, retrieved March 2016 from http://volcano.si.edu/). Comparison of these snow measurements to previous seasonal snow measurement campaigns (e.g., Davidson et al. 1993; Toom-Sauntry and Barrie, 2002) showed this fall peak in $SO_4^{2-}$ snow concentration and flux to be unusual. This further supports a non-seasonal event such as a volcanic eruption as a major contributor to Factor 7, sulphate. These volcanic sources are shown on the source region plot for Factor 7 sulphate as well as the smoking hills for context (Figure 3). Although none of these locations appear to have high associated residence

times within the ten-day back trajectory analysis, they appear closer to the identified source regions than major industrial activities farther south. Given the heat and velocity of a volcanic emission, the near-surface restriction applied to the FLEXPART trajectories to identify likely source regions may not be appropriate for Factor 7. Supplemental section S2.2 provides potential source/influence region plots for peak fall periods associated with Factor 7 for a larger range of source altitudes: 0 to 10 km above ground level. These plots do show potential influence from Bárðarbunga in Iceland.

The coincident enhancement of MSA production within the ice-free biogenically-active Arctic Ocean in the fall would explain why MSA was also found to be loaded onto this factor. Photochemical $SO_4^{2-}$ apportionment sources have been observed in some previous apportionment studies, though not as commonly as other factors (e.g., Sirois and Barrie, 1999). Significant volcanic influences on Arctic aerosol have also been noted by previous apportionment studies (VanCuren et al., 2012).

Previous analysis of snow and atmospheric samples over this campaign (Macdonald et al, 2017) found $SO_4^{2-}$ to exhibit an enhanced deposition velocity relative to BC, especially during the warm fall months. Typical internally-mixed anthropogenic particulate $SO_4^{2-}$ and BC would be expected to exhibit similar deposition velocities; thus, this discrepancy supported a distinct fall source of $SO_4^{2-}$ which was more readily scavenged/deposited than BC. It is possible that heightened scavenging of volcanic $SO_2$ emissions in the warmer fall resulted in this seasonal trend and the identification of a separate non-anthropogenic $SO_4^{2-}$ factor.

Within a six-factor solution, Factors 6 and 7 were essentially combined into a single factor. This combined factor did not exhibit a clear distinct source region, nor was it easily interpretable. The six-factor solution also did not predict the observed distinct fall peak in $SO_4^{2-}$ (as shown in supplemental Figure S7). Thus, the use of a seven-factor solution vastly improved $SO_4^{2-}$ apportionment for this campaign.

### 3.3 Overall Apportionment

Considering Figure 3 and Table 2, the apportioned factors can be split into two groups by potential influence/source regions: those dominated by anthropogenic sources and those by natural sources. All factors showed a significant influence from Arctic regions since all trajectories were initialized at Alert, but three factors showed heightened influence from areas outside of the Arctic: Factor 3 black carbon, Factor 5 nitrate, and Factor 6 non-crustal metals. All three were observed to have potential areas of influence extending south into Eurasia, up to and below 45 °N. Although each factor is potentially an amalgamation of several co-emitted or co-aligned sources, per the discussion above, the composition and peak periods of Factor 3, BC, and Factor 6, non-crustal metals, suggest they are primarily anthropogenic in origin. While Factor 5 appears to represent the distinct $NO_3^-$ atmospheric chemistry, the precursors to these reactions may also be anthropogenic in origin. Factor 1, sea salt, also showed a large influence area within the northern Eurasia (per Table 2); however, it is believed that this represents influence from the coast during ice-free periods. Thus, Factor 1 sea salt, Factor 2 crustal metals, Factor 4 carboxylic acids, and Factor 7 sulphate, all appear to be dominated by influences/sources north of 65 °N. Per the discussion above, these factors all appear to have largely natural sources, although additional evidence on the identity of Factor 4 and 7 in particular is warranted. Based on these identifications the rough proportion of each analyte apportioned to factors which most closely resemble anthropogenic sources (Factor 3 BC, Factor 5 nitrate, and Factor 6 non-crustal metals) or natural sources (Factor 1 sea salt, Factor 2 crustal, Factor 4 carboxylic acid, and Factor 7 sulphate) can be estimated. Table 3 provides a summary of this classification.

Table 3 indicates that $NO_3^-$, BC, Sb, and Pb are likely all dominated by anthropogenic sources. In contrast, $Mg^{2+}$, Se, Al, $Na^+$, V, formate, acetate, $Cl^-$, $SO_4^{2-}$, $NH_4^+$, and $K^+$ are all likely dominated by natural sources. The total apportionment of As, $C_2O_4^{2-}$, MSA, Br-, and Cu to anthropogenic and natural-resembling factors was found to be uncertain based on the bootstrapping analysis. Most notable in this analysis was the apportionment of BC, $SO_4^{2-}$, V, and Se. While the loading of BC is known to vary between anthropogenic and natural sources by location and season, typically $SO_4^{2-}$, V, and Se would be expected to be primarily anthropogenic in origin. Figure 4 summarizes the apportionment of BC, $SO_4^{2-}$, and V over the campaign. The apportionment of Se was similar to that of V.

**Table 3: Overview of analyte apportionment.**

| Factor | 1 Sea Salt | 2 Crustal Metals | 3 BC | 4 Carboxylic Acids | 5 Nitrate | 6 Non-crustal Metals | 7 Sulphate | Total Loading by Most-Closely Resembled Source Type | | | |
|---|---|---|---|---|---|---|---|---|---|---|---|
| | | | | | | | | Primarily Anthropogenic (Factors 3,5*,6) | | Primarily Natural (Factors 1,2,4*,7*) | |
| Analyte | Loading (mass/mass) | | | | | | | | | | |
| BC | 5% | 0% | 66% | 17% | 0% | 8% | 5% | 73% | [60-124%] | 27% | [3-67%] |
| MSA | 5% | 0% | 10% | 12% | 27% | 4% | 42% | 41% | [6-128%] | 59% | [26-176%] |
| ACE | 0% | 5% | 10% | 79% | 0% | 0% | 6% | 10% | [8-75%] | 90% | [51-155%] |
| FOR | 0% | 10% | 0% | 80% | 1% | 8% | 0% | 9% | [3-69%] | 91% | [50-151%] |
| $Cl^-$ | 79% | 3% | 7% | 2% | 2% | 2% | 5% | 11% | [7-40%] | 89% | [69-110%] |
| $Br^-$ | 34% | 0% | 0% | 26% | 23% | 15% | 4% | 37% | [2-108%] | 63% | [30-138%] |
| $NO_3^-$ | 0% | 10% | 4% | 0% | 86% | 0% | 0% | 90% | [64-111%] | 10% | [4-58%] |
| $SO_4^{2-}$ | 10% | 4% | 5% | 0% | 4% | 9% | 68% | 18% | [14-76%] | 82% | [62-141%] |
| $C_2O_4^{2-}$ | 27% | 8% | 9% | 8% | 14% | 22% | 12% | 45% | [27-83%] | 55% | [30-116%] |
| $Na^+$ | 79% | 4% | 0% | 2% | 0% | 7% | 9% | 7% | [4-41%] | 93% | [73-118%] |
| $NH_4^+$ | 15% | 2% | 17% | 47% | 5% | 5% | 8% | 28% | [25-80%] | 72% | [41-111%] |
| $K^+$ | 38% | 10% | 0% | 4% | 20% | 10% | 19% | 29% | [8-93%] | 71% | [36-143%] |
| $Mg^{2+}$ | 43% | 34% | 1% | 5% | 0% | 0% | 17% | 1% | [1-37%] | 99% | [76-126%] |
| Al | 2% | 84% | 0% | 0% | 3% | 3% | 7% | 7% | [2-43%] | 93% | [70-136%] |
| V | 2% | 84% | 1% | 1% | 3% | 5% | 5% | 9% | [3-47%] | 91% | [68-130%] |
| Cu | 6% | 48% | 0% | 0% | 7% | 28% | 11% | 35% | [19-82%] | 65% | [32-143%] |
| As | 5% | 44% | 7% | 0% | 0% | 44% | 0% | 52% | [15-109%] | 48% | [38-106%] |
| Se | 0% | 81% | 2% | 1% | 0% | 3% | 12% | 6% | [2-44%] | 94% | [69-141%] |
| Sb | 0% | 0% | 4% | 18% | 1% | 60% | 17% | 64% | [42-126%] | 36% | [11-99%] |
| Pb | 4% | 25% | 8% | 8% | 0% | 53% | 2% | 61% | [35-95%] | 39% | [27-108%] |

**Notes:** Factors classified based on available evidence as most closely resembling anthropogenic sources (Factor 3, 5, and 6) or most closely resembling natural sources (Factor 1, 2, 4, and 7); however, all factors likely represent an amalgamation of different sources. * Denotes particular uncertainty in classification. Loading described as: optimal solution [25th - 75th bootstrapping].

5 As discussed above, Arctic BC is often considered to be primarily anthropogenic in origin over the winter season; however, there is some contradictory evidence. It would appear that the sources of BC to Arctic snow vary by location, season, and year. Figure 4 shows that snow BC in this study was dominated by Factor 3, believed to be predominantly anthropogenic in origin. Only about 17% of BC was loaded onto the factor most resembling biomass burning, Factor 4 carboxylic acid, similar to the findings of previous modelling and composition-based apportionment estimates for particulate matter (Slater et al., 2002;

10 Flanner et al., 2007; Skeie et al., 2011; Wang et al., 2011; Yttri et al., 2011; Yttri et al., 2014). The dominant factor for BC varied over the campaign: Factor 3 (BC) was dominant from November through April, but Factors 4 carboxylic acids and Factor 7 sulphate showed larger contributions in the fall and spring. However, given the low levels of BC observed over fall and spring, the absolute contributions of Factor 4 and 7 were small and susceptible to significant uncertainty. The portions of BC assigned to Factors 1 sea salt and Factor 2 crustal metals likely represent a regional background level of BC and therefore

15 are likely the combined product of both anthropogenic and natural emissions.

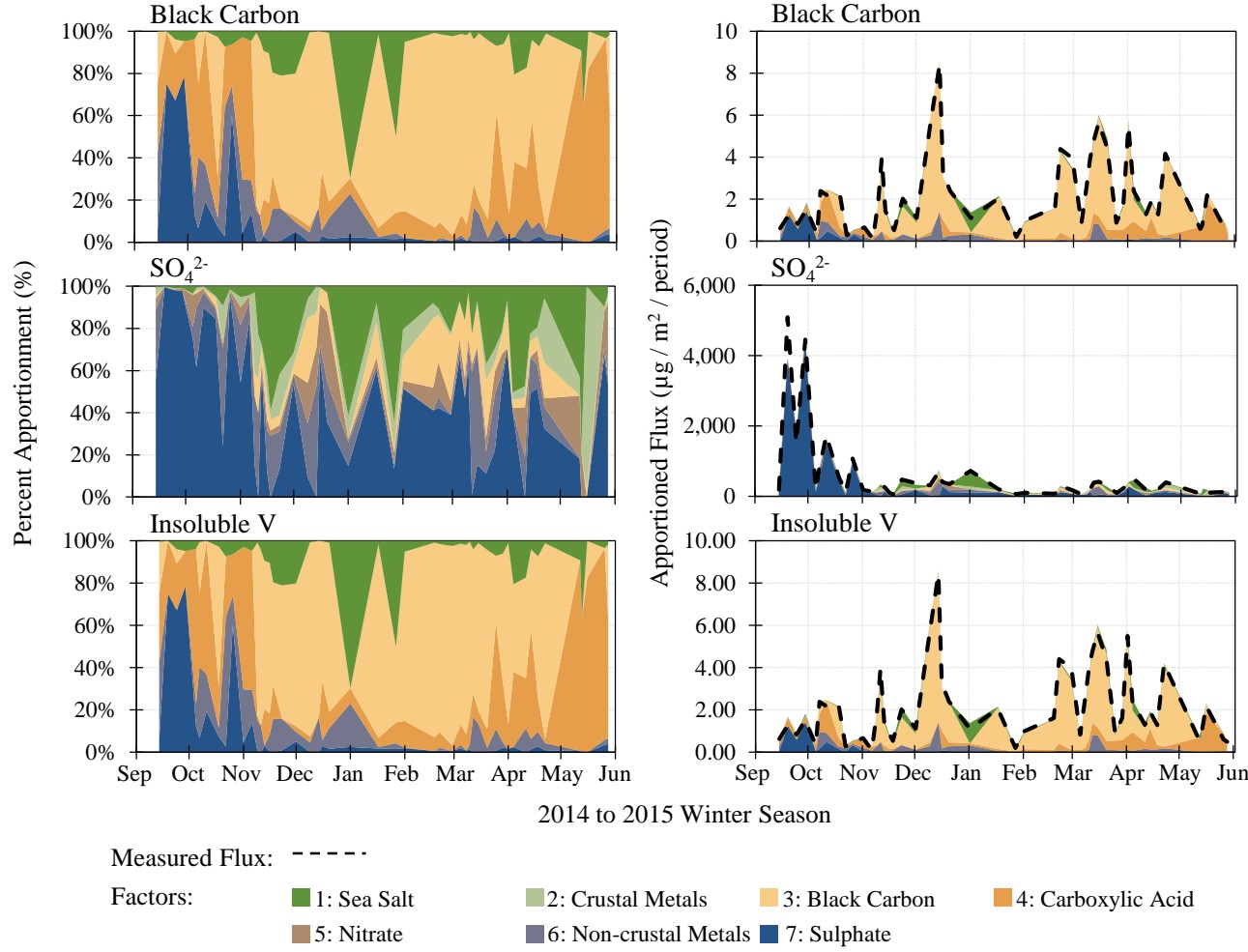

**Figure 4: Percent and total apportionment of BC, $SO_4^{2-}$, and V in snow over the 2014/15 campaign.**

The $SO_4^{2-}$ signal was dominated by a fall peak primarily loaded onto Factor 7. While additional evidence is required to
5  corroborate the identity of this factor, the coincidental eruption of Bárðarbunga in Iceland may suggest a significant volcanic
source during the fall of this campaign. The absolute flux of $SO_4^{2-}$ in the winter and spring was relatively small compared to
the fall peak and mostly comprised of Factor 1 sea salt and Factor 7 sulphate. However, episodic peaks in Factor 3 BC and
Factor 6 non-crustal metals suggest influence from anthropogenic plumes. Excluding the September/October peak, the $SO_4^{2-}$
loading is approximately 39 % anthropogenic (12% Factor 3 BC, 9% Factor 5 nitrate, and 18% Factor 6 non-crustal metals)
10  and 61% natural (23% Factor 1 sea salt, 10% Factor 2 crustal metals, 0% Factor 4 carboxylic acid, and 28% Factor 7 sulphate).
Thus, factors considered anthropogenic account for about 50% of the non-sea salt $SO_4^{2-}$ signal over this period. If Factor 7
were miss-identified as natural, then approximately 67% of $SO_4^{2-}$ or 87% of non-sea salt $SO_4^{2-}$ over the winter/spring would
be considered likely anthropogenic in origin.

Both V and Se are typically considered to be tracers of anthropogenic activity, specifically oil and coal combustion (These factors were

identified based on composition, seasonaggan, 1953; Rahn, 1971; Berg, Røyset, and Steinnes, 1994; Laing et al., 2014). However, the dominant sources of V and Se observed in this study was soil in Factor 2 crustal metals. The loading of these

metals relative to Al appeared consistent with the range previously observed by others in soils; also, the raw concentration measurements of these metals showed high correlation with Al. Thus, the apportionment of these metals to primarily natural sources is considered reasonable. As shown in Figure 4, V shows episodic peaks in Factor 5 nitrate, Factor 6 non-crustal metals, and Factor 7 sulphate. Both Factor 5 and 6 are believed to be predominantly anthropogenic in origin; thus these peaks may represent episodic plumes from oil/coal burning activities. However, as shown in Table 3, the total loading of V and Se

to these factors are low.

In general, the apportioned analytes differed in how exclusively they were attributed. Some analytes were found to be predominantly loaded onto a single factor: BC, sea salt, and crustal metals. This may indicate that much of the mass of these analytes exist in externally mixed particles, or internally mixed with a relatively small coating mass. In contrast, other analytes were found to be loaded more evenly onto several factors: MSA, $Br^-$, $K^+$, and $C_2O_4^{2-}$. Thus, these analytes may exist primarily

as internally mixed particles or gas-phase compounds. This assessment is in agreement with previous explorations in the deposition characteristics of this data (Macdonald et al., 2017).

## 4 Conclusions

The Arctic climate has undergone significant climate change over recent decades and any effort to control and mitigate these changes requires improved understanding of the source contributing to the Arctic snow burden. The data presented here

represents an unprecedented campaign of temporally-refined and broadly speciated snow samples which is the first of its kind to be applied to a detail source apportionment analysis. Positive matrix factorization of the snow measurements was found to resolve seven factors with good solution diagnostics, interpretability, and agreement with measured values. These factors were identified based on composition, seasonal contribution, and FLEXPART-predicted major source regions: sea salt, crustal metals, black carbon, carboxylic acids, nitrate, non-crustal metals, and sulphate. Based on possible factor identification, BC

apportionment was found to load 73% m/m of the total flux onto factors considered to be primarily anthropogenic in origin; however, the lower levels of BC in the fall and spring were largely associated with factors which might be associated with North American biomass burning. These BC apportionment results reiterate the importance in understanding the variation in BC sources by year, location and season. In contrast, $SO_4^{2-}$, V, and Se were only attributed to factors resembling anthropogenic sources by 18%, 9%, and 6% m/m, respectively. The $SO_4^{2-}$ signal was dominated by a fall peak with limited BC loading. Based

on the coincidental eruption of a volcano in Iceland and the lack of anthropogenic tracers, this peak was believed to be predominantly natural in origin. This result may indicate the importance of high volcanic activity years. The low anthropogenic V signal was due to significant loading of V onto a crustal source. The ratios of V and Se to Al in this factor were fairly

consistent with the typical range seen in soils; furthermore, the raw measurements of both metals showed high correlation with Al. The anthropogenic signal of V and Se was largely attributed to a factor dominated by non-crustal metals which was believed to represent mixed Eurasian anthropogenic emissions. Comparison of these results to a previous analysis of the deposition characteristics of this data highlighted the importance of relative deposition velocity and mixing state in the apportionment of analytes in snow. Future analyses of Arctic snow would be required to fully understand these complexities.

## Author Contribution

Organization of the snow collection campaign was led by S. Sharma with the assistance of A. Platt and collection by M. Elsasser. Snow analyses were completed by J. McConnell, N. Chellman, D. Toom, L. Huang, and K. Macdonald with the assistance of A. Chivulescu, Y. Lei, and C.-H. Jeong. Ambient atmospheric monitoring was completed by D. Toom and R. Leaitch. FLEXPART simulations were completed by H. Bozem and D. Kunkel with data analysis assisted by K. Macdonald. PMF analysis was completed by K. Macdonald with input on interpretation from all authors. Dr. G. Evans and J. Abbatt provided oversight for the project, including input on the manuscript.

## Competing interests

The authors declare that they have no conflict of interest.

## Acknowledgements

Funding of this study was provided as part of the Network on Climate and Aerosols Research (NETCARE), Natural Science and Engineering Research Council of Canada (NSERC), the government of Ontario through the Ontario Graduate Scholarship (OGS), and Environment and Climate Change Canada. This project would not have been possible without the collaboration of many skilled individuals, including Allan K. Bertram and Sarah Hanna at the University of British Columbia and Catherine Philips-Smith at the University of Toronto.

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
