# Peer review of "Temporally-Delineated Sources of Major Chemical Species in High Arctic Snow"

_Atmospheric Chemistry and Physics, 2017_

## Referee Comment (RC1) · Anonymous Referee #1 · 7 Sep 2017

This manuscript is the second to report on the results of 9-10 month long campaign (September to June) characterizing the chemical composition of fresh snow sampled at Alert. The first paper presented the data and compared it to simultaneous measurements of aerosol composition to assess the efficiency of air to snow deposition for the different analytes. Here the focus is application of PMF and the FLEXPART transport modeling tool to assess source regions for the various chemical compounds measured in the snow.

This is a solid piece of work, though I feel that the manuscript is less accessible than it could be (more on that below). I also suggest that the authors should consider changing the emphasis in several places in the discussion, to better reflect a lot of other recent (and also pioneering) work on related topics. A very good example of this arises

as early as the abstract, where the finding that BC in the high Arctic during winter is dominantly from anthropogenic sources (fossil fuel combustion) and not biomass burning is highlighted. In section 3.2.2 their analysis refines this even more and points to sources in Eurasia for nearly all of this anthropogenic BC. To me, this is basically rediscovering some of the very early findings from a host of "Arctic Haze" investigations initiated in the 1970s which documented that the Haze was largely pollution, it was significantly absorbing due to BC, and much of it came from relatively high latitudes in Europe and Russia. Authors note that their work is focused on snow rather than aerosol, yet they explicitly assert that the snow is providing constraint on aerosol sources, so this "finding" is reassuring but perhaps not so exciting as to merit being the only factor from the PMF to be called out in the abstract. This statement about BC in the abstract notes that it is a "light-absorbing compound critical to the Arctic radiative balance" which is certainly true. However, the AMAP, 2015 assessment (cited frequently in this manuscript) points out that a suite of CTMs all agree that Asian sources dominate the atmospheric burden and climatic impact of BC in the Arctic. Most likely this apparent discrepancy is due to the highly stratified Arctic winter time troposphere, allowing Eurasian BC sources to be dominant in lower levels (sampled at surface aerosol sites and scavenged by mid- to low-level clouds) while Asian BC is at higher altitudes. In any case, I find the present result that essentially no Asian BC gets to Alert within 10 days more interesting than seeing very little biomass burning smoke in the high Arctic during winter.

A very interesting finding in this work is the lack of a strong anthropogenic sulfate signal. Arctic Haze "comprises a varying mixture of sulfate and particulate matter and, to a lesser extent, ammonium, nitrate, dust, and black carbon (e.g., Li and Barrie, 1993; Quinn et al., 2002)" (Quote from chapter 4 of AMAP, 2006; another work cited several times in this manuscript. This statement is also repeated nearly verbatim on page 2 lines 10-11 of this manuscript.) This may reflect imperfect air-snow transfer of a defining characteristic of the Arctic winter-time troposphere, greatly enhanced sulfate, or possibly strong impact from volcanic sources in this particular year (suggested by the

authors, but not very convincingly). Critically assessing air to snow transfer of sulfate would provide a nice link to the first paper in this series. However, the missing Arctic Haze sulfate signal could also reflect problems arising from sampling fresh snow from elevated snow tables (see more on this in first detailed comment below).

One final example of a finding that is perhaps misinterpreted or at least somewhat misrepresented is the attribution of PMF factor 2 to local dust. V, Se, and As are generally considered to be dominated by anthropogenic emissions, and in fact the authors point this out in their later discussion of factor 6. In particular, finding V to be enriched in Arctic Haze caused Ken Rahn to reassess, and basically refute (Rahn et al. 1985 in Atmos. Environ., see also AMAP, 2006, chapter 4), his own early suggestion that the haze was mostly dust from Asia (Rahn et al., 1977 in Nature). Mosher et al., 1993 used V to show that emissions from the generators at the DYE 3 radar station probably had a subtle but persistent impact on aerosol measurements made during the DGASP campaign. (Pretty well established that V is a tracer of oil combustion, in fact the authors point this out in discussion of factor 7.) Given the correlation between factor 2 and winds from the main station at Alert, it would seem plausible that local pollution, and not just local dust, is part of this factor.

Regarding comment about accessibility of the manuscript, the very detailed description of PMF in section 2.4.1 and section 3.1 describing how 7 factors were ultimately selected is too lengthy for a journal like ACP, especially considering that the algorithm is publicly available and presumably well described in EPA documents and Norris et al., 2014. Material in the supplemental showing the changes as additional factors are considered is well done, but not distracting to someone reading the paper who may be less interested in statistical details.

Detailed comments. The first paragraph of section 2.1 probably needs to be expanded to provide a few additional details about sampling and data screening. In particular, in Macdonald, 2017 the chemical fluxes in January and February were excluded in all analyses due to indications that the snow tables suffered extreme undercatch during

high winds in mid winter. However, in this manuscript these data are retained, the PMF is conducted on "flux per snowfall event" rather than concentration or flux per day, and spikes in several of the factors during January and February were used to support attribution of the factor to source. Authors need to justify this pretty large change in assessment of data quality (or stick with original decision and leave mid winter out of the PMF). As noted above, I wonder if low fluxes due to snow undercatch obscured the expected winter peak in sulfate flux.

Figure 3 probably needs to be modified, given its central role in attributing factors to likely sources. All 7 panels share a lot of similarities that tend to draw the eye as, or even more, strongly than small differences pointed out in the text in section 3.2. Probably the biggest problem is the bulleye very close to Alert in all of the panels. This is largely a geometric artifact reflecting that every particle released from the receptor site has to pass through a very small number of cells surrounding that site. I am pretty sure that Stohl and/or Burkhart have recognized this issue and have a recommended weighting scheme that reduces this bias (lower weights for cells closer to release site). Another minor point is that the green triangles and square in the panel for factor 7 are very hard to find (especially the Smoking Hills square). And the label under color bar should be Residence Time (not Residential), and there has to be some huge multiplier on the scale (max is not just 30 seconds)

Editorial comments by page/line number

1/23 AMAP 2011 was updated in 2017, probably should cite that report

2/6-8 Not sure how the concluding phrase about snow as a critical reservoir logically follows the first part of this sentence.

2/8-17 Given the vast literature on Arctic Haze, it is unclear how the references in this section were selected. Personally, I would like to see some of the very early work cited. At a minimum, indicate that AMAP, 2006 is a review paper and readers should see references cited therein.

3/4 The last phrase after the comma is very much a matter of personal opinion. I suggest ending sentence with a period after flux (see first detailed comment above).

3/20-21 Reword this to make argument more clear, and possibly consider different wording for "under-exaggerate". Are you saying that you tossed BDL samples to make the S/N higher than it probably should have been?

5/32 to 6/6 Is this needed? Results from PCA are not shown, and appear to be mentioned in passing just once more in the manuscript (page 8, line 8)

6/19 residential-→residence

9/1 Enhancement of Mg above the SS ratio by a factor of 1.6 is a big difference that would suggest an additional Mg source. Same is true for SO4, but excess is expected.

9/14-15 The residence time plot suggests that the middle of the GrIS is a stronger source for this factor than Norwegian Sea or North Atlantic, probably partly due to geometric artifact mentioned earlier.

10/Figure 1 Please explain what the bars on this plot are showing more clearly. What is the time component indicated by "/period" ?

12/Figure 3 Why not label the panels by source name rather than factor #?

13/29 There have been a lot of papers on emissions from fires (lab, prescribed, and wild) since 2009. Liu et al., 2017 in JGR maybe most recent. This one does not include BC, but provides access to many of the papers between 2009 and 2017. 14/1-16 Hirdman et al. 2010 (2 papers, in ACP) and Stohl et al 2006 (JGR) have shown similar. They probably should be cited.

16/1 delete "both"

16/3 delete "to"

16/3-4 There have been a lot of papers on emissions from fires (lab, prescribed, and

wild) since 2009. Liu et al., 2017 in JGR maybe most recent. This one does not include BC, but provides access to many of the papers between 2009 and 2017.

17/11 why not say "via N2O5 hydrolysis in the aerosol phase" instead of "NO3-radical chemistry"?

18/14 Laing et al. 2014 is not original source of this fact, Rahn probably closer, but maybe even he used someone else's earlier work

18/20-21 Fact that FLEXPART rarely reaches any of these volcanoes is a little problematic.

21/7 seasonally-→seasonal

---

## Referee Comment (RC2) · Anonymous Referee #2 · 25 Sep 2017

Macdonald et al describe the results of positive matrix factorization of snow chemical composition measurement data from Alert, Nunavut in order to determine the prominent sources influencing the snow composition. Given changing Arctic source emissions with sea ice loss and increasing development, this is an important topic. A thorough description of the data analysis is provided. My main concerns, described below, surround the discussion of the results.

The main result highlighted in the abstract and conclusions is that the BC is primarily from fossil fuel burning, rather than biomass burning influence. This is not surprising since the study focuses on snow samples collected from Sept. 14, 2015 to Jun. 1, 2015, outside of the main summertime wildfire period. In several places in the paper (last paragraph of Section 3.2.3, part of Sec. 3.3, and P21 L 11-14), it is stated that

[Figure]

these results "disagree" with previous snow chemical composition measurements that showed greater biomass burning influence, proving "contradicting snow BC apportionment findings". The authors do note the influence of seasonality and changes in annual wildfire frequency and severity on contributions of biomass burning BC. However, because the references that the authors are comparing to correspond to different times and locations, a simple comparison of the percentages of biomass burning vs fossil fuel influence is not appropriate (e.g. Table 2), without an in-depth analysis of fire locations, frequency, and timing, as well as air mass trajectories associated with the various sampling sites. I would expect that the contribution of biomass burning vs fossil fuel likely depends on the site, season, and year. Therefore, I suggest revising the discussions and comparisons to provide these results as another study that points to the variability in BC source contributions, rather than suggesting that they "disagree with" or "contradict" previous results, which gives the idea of invalidating previous work, which instead may simply be different due to different timing and location. As part of this revision of the discussion, I suggest removing Table 2, or if the authors feel strongly about keeping this comparison, then information about timing, location(s), and wildfire influence (from fire maps and air mass trajectory analysis, presumably, or statements from previous papers) should be included. In addition, a more thorough literature search is needed if the authors mean for this to be a comprehensive comparison.

This is a complementary paper to the recent Macdonald et al (2017) ACP manuscript that describes the deposition of the same chemical species to the snowpack, with snow mixing ratios and fluxes of these species described. In that paper, Figure 1 shows time series over the same period of Sept 2014 to Jun 2015 for the following "key analytes" (as described in that paper), grouped according to time series correlations: Black carbon, methanesulfonate, $C_2O_4^{2-}$ & $NH_4^+$, sea salt, NSS-sulfate, nitrate, NSS-$K^+$ & NSS-$Br^-$, and crustal metals; this is quite similar to the time series of the 7 factors (salt, dust, BC, carboxylic acids, nitrate, metals, and sulfate) in Figure 2 of the current paper. Despite this overlap, little discussion was included in the previous manuscript regarding likely sources.

The authors are encouraged to do a more thorough literature search for previous Alert snow, aerosol, and trace gas studies that likely will support their source apportionment findings and provide evidence for greater certainty for source identification. Some appropriate papers (not meant to be comprehensive) are noted below for discussion of specific factors. While not temporally resolved, Krnavek et al 2012 (Atmos. Environ.) provide a detailed source apportionment of marine, terrestrial, and atmospheric influences on Arctic surface snow composition. Most notably, the authors do not cite or compare to Toom-Sauntry and Barrie (2002, Atmos. Environ) who previously collected weekly snow samples at Alert from 1990 to 1994 and measured inorganic and organic ions; this paper is highly relevant to the current work!

Major comments:

Abstract: Currently, only two results are noted here – the names of the source factors and the fossil fuel source of the BC. Can additional results associated with other factors be mentioned here to highlight this work? Also, please be consistent between the factor names here and throughout the text (e.g. this says "regional dust", but later it is discussed that the dust is likely local).

Section 3.1.1 Factor 1 (Marine Sea Salt): Is there seasonal dependence to the Br-enrichment factor? There is well-known multiphase bromine chemistry that occurs in the Arctic in the spring (see Simpson et al. 2007, ACP). Hara et al (2002, J. Geophys. Res.) conducted a detailed examination of Br- enrichments in Arctic aerosols and may be useful to consider for this work. A neutralization ratio of 0.8 is stated as neutral; what is the uncertainty associated with the calculated ratio? The discussion of the potential sea salt sources is muddled with respect to local vs far away sources and should be clarified, with improved flow in discussing the possibilities. Note that recent work has suggested that aerosols are not produced from frost flowers (Yang et al 2017, ACP; Roscoe et al 2011, J. Geophys. Res.). Were there are open leads upwind of the field site, such that open water was closer to the site? May et al. (2016, J. Geophys. Res.) pointed to sea salt production from leads in the fall-spring.

Section 3.2.3 Factor 3 (BC): The authors should consider the work of Doherty et al (2010, ACP), who measured light-absorbing impurities in ~1200 snow samples across the Arctic. Dou et al (2012, ACP) previously compared measured snow BC to simulations of the spatial distribution of snow BC using the GISS-PUCCINI model. Recently, Barrett et al (2015, Environ. Sci. Technol.) used radiocarbon tracers to determine elemental carbon source apportionment between modern and fossil fuel carbon at Barrow, AK; perhaps some discussion in that work may be helpful here.

Section 3.2.4 Factor 4 (Carboxylic Acids): In the authors' consideration of carboxylic acid sources, they should consult the work of Narukawa et al (2002, Atmos. Environ.) who measured aerosol and surface snowpack dicarboxylic acids at Alert in Feb and April-May 2000. Dibb and Arsenault (2002, Atmos. Environ.) examine snow as a source of acetic and formic acids.

Section 3.2.5 Factor 5 (Nitrate): The authors cite Morin et al (2008) and Fibiger et al (2016) for nitrate cycling associated with the snowpack. However, Fibiger et al (2016) is a study at Summit, Greenland. There are other appropriate studies at Alert that should be considered in the context of the current work – for example, Ianniello et al (2002, Atmos. Environ) and Beine et al (2002, Atmos. Environ.).

Section 3.2.7 Factor 7 (Sulfate): In considering the main sources of snow sulfate, the authors should consult the work of Norman et al (1999, J. Geophys. Res.) who used sulfur isotopes to determine seasonal aerosol sulfate sources at Alert from July 1993 to Sept. 1994. The authors note that several volcanoes were active over the 2014-2015 season. This factor peaks in the early fall; does this coincide with the volcano activity and associated air mass trajectories (FLEXPART analysis)? Reorganize this section so that there is a clear flow of discussion – currently the authors go back and forth between multiple potential sources. For example, L20-21 and 27-29 seem to be somewhat contradictory as written. L6-7 on P19 seems to be tacked on and should be integrated.

Section 3.3: This section is labeled as "Overall Apportionment", but it is really primarily a discussion of how BC is apportioned between the factors. It may be useful to rename the title of this section, or reorganize and revise the section to make it more evenly about all of the factors. I would suggest a paragraph break at L21, with some reorganization between the two paragraphs. The authors point to mixing state of the particles potentially being important (L23-25), and this could be strengthened by citing previous Arctic studies (e.g. Weinbrunch et al 2012, Atmos. Environ.).

Table 3: It would be useful to integrate these results into the prior factor discussions (section 3.2).

Conclusions: The conclusions are very general, with limited discussion of any factor or analyte other than BC. There is an opportunity here to discuss other factors and analytes, particularly with respect to how they may change in the future, or with respect to uncertainties that should be examined in future work.

Minor Comments & Technical Corrections:

P1 L19, P8 L13, & in other locations: Please clarify text to describe the units used for calculating the percentage. I assume for BC that you are calculating the % based on mass conc? For Na+, for example, are you reporting the fraction of Na+ measured in the snow that was apportioned to the first factor? This isn't currently clear and could be worried more clearly throughout the manuscript where percentages are used.

P1 L19: Fix phrasing/sentence structure as snow is not a light-absorbing compound.

P2 L7-10 & L14-15, P15 L15, & P17 L8-9: Provide references.

P2 L15: Please clarify the phrase "less prone to the ambiguities introduced by snow-pack collection".

P2 L27-28: Mention measurements data here – otherwise it sounds like the study includes only PMF and air mass modeling.

P7 L3: Is this supposed to be 59 samples (based on P3 L2)?

P7 L4: Clarify wording that you are discussing analyte concentrations and fluxes.

P7 L13-14: This discussion is not intuitive and could be clarified further. Can we learn about processes from these differences?

P7 L17-33, P8 L5-9, & Table 1: I suggest moving these paragraphs to the methods and supplementary information, as they discuss how the authors decided to use seven factors and do not discuss science. The section is also difficult to follow without in-depth knowledge of the method, and without referring back to the methods section frequently. Similarly, I suggest moving Table 1 to the supplementary information.

P8 L14-15: These sentences are redundant.

P9 L1-2: "Compound(s)" should be "ion(s)" here. Also, what are the uncertainties in the enrichment ratios? (These errors should be stated for all enrichment ratios reported in this manuscript.)

Figure 1: Remove "(point)" and "(bar)" on the y axes, as this is already shown in the legend, and "bar" is a unit of pressure.

Figure 1 caption: Provide further description of how to interpret the figure for improved clarity, particularly for those not familiar with PMF.

Figure 2 caption: What are the traces normalized to (themselves, other factors?)? What are the units? This caption is not clear.

P11 L9: Provide the calculated ratio in parentheses for context.

P13 L32: The neutralization equation is provided on P9, but it is not clear if the same equation is used for the calculation here and elsewhere in the paper.

Table 3: While there is a footnote defining "Southern Oceans", I suggest renaming to Atlantic & Pacific Oceans, since "Southern Ocean" is a phrase typically referring to

near the Antarctic.

Figure 4: The abbreviation "Cbx. Ac." In the legend is not immediately obvious; I suggested writing out "carboxylic acids" on two lines instead for improved clarity.

---

## Referee Comment (RC3) · Anonymous Referee #3 · 23 Oct 2017

Review for Atom. Chem. Phys. Discuss. Temporally-Delineated Source of Major Chemical Species in High Arctic Snow General review: The paper provides apportionment of chemical components in high Arctic snow, which is of interest. Some of the interpretation of source region and emission source connected to the PMF factors was not sufficiently supported and seemed stretched; this was particularly true for the discussion for the sulfate factor and the attribution of V, As and Se to dust/crustal materials in the dust factor. Improved consistency is needed for naming across the text, figures, and tables. I agree with comments provided by the previous referees.

Detailed comments:

P 3 Ln 5-7. You need to give a bit more detail here, regardless of whether you are following previous protocol as this paper needs to be able to stand alone.. How are

these melted? How is the filtration accomplished? What is the storage protocol? How are the blanks?

P6 Ln 17: Please make this more explicit, especially for ones where the is temporal overlap in the peak concentration of the factor.

P8 Ln12: Please make all factor names consistent: sea salt/marine sea salt/marine factor, choose one and use for all tables, text and figures.

P9 ln13: You should be able to find the ice extent for these specific time periods for the locations mentioned. Also, based on the heat map in Figure 3 for Factor 1 (you should really include the Factor names here as well, as it is difficult to keep track of which factor is which across a couple figures), the longest residential time is north of Greenland and Siberia – are these areas open water in January 2015? Wouldn't the open water have to have been close to the site for the correlation to local wind speed be relevant for sea spray sourcing?

Figure 1: clarify whether these are soluble, insoluble or total metals.

P11 ln3: make all factor names consistent throughout the manuscript: crustal metals vs dust. Also, the high contribution of V, As and Se might indicate anthropogenic pollution (i.e. coal or heavy oil combustion) not just "dust".

Figure 3: the cyan diamonds and green triangles are very difficult to see.

P14 ln10: for Russian BC sources, there have been two new studies in the last year that should be included here and incorporated into the discussion: Evans, Meredydd, Nazar Kholod, Teresa Kuklinski, Artur Denysenko, Steven J. Smith, Aaron Staniszewski, Wei Min Hao, Liang Liu, and Tami C. Bond. "Black carbon emissions in Russia: A critical review." Atmospheric Environment (2017). Winiger, Patrik, August Andersson, Sabine Eckhardt, Andreas Stohl, Igor P. Semiletov, Oleg V. Dudarev, Alexander Charkin et al. "Siberian Arctic black carbon sources constrained by model and observation." Proceedings of the National Academy of Sciences(2017): 201613401.

P14: for detailed comparison with previous high Arctic snow apportionment studies, do also take into account more of the potential impact of Arctic location. The Hegg studies were quite different in the study design, representing PMF of a large number of Arctic sites as opposed to PMF at a single Arctic site.

Table 2: include location of the studies. The location is very relevant in terms of understanding BB impact across the Arctic. For the apportionment/co-variance (again, use the same terms in the text and tables to avoid confusion), include types of species used in the modeling for BC apportionment.

Pg16 ln 1: I think this sentence has been truncated ". . .linked with both biomass burning plumes. . ." and?

P17 ln13: where are source areas shown in Figure 2?

P17 ln15-16: It's not clear how this factor coincides with increased transport over the ice-free Norwegian Sea and northern Atlantic. Remove unless you can support

P18 ln 20-21: the Flexpart in Figure 3 does not seem to match with the assignation of sulfate to volcanoes and the Smoking Hills.

P18-19: the explanation for the sulfate factor was a bit forced to match volcanism. If the metals factor was combined with sulfate in the six factor solution, it would seem that would indicate an anthropogenic source. When comparing to the connected Macdonald paper, the co-variance of sulfate and MSA (or MS, as it was called in the previous paper), might be spurious as MSA is only high in the early part of the campaign.

Figure 4: use the same naming for factors across all figures, text and tables. The abbreviation is difficult here.

P21 ln 13: again, take location into account for comparison with other Arctic BC studies.

---

## Author Comment (AC1) · 16 Dec 2017

Temporally-Delineated Sources of Major Chemical Species in High Arctic Snow – Response to Anonymous Referee #1

Referee comments received and published: 7 September 2017 (quoted below)

We would like to thank Referee #1 for their detailed comments and discussion. We greatly appreciate the care with which the referee has reviewed this manuscript and the improvements gained through their insight. Response to Referee Discussion Referee Comment: This manuscript is the second to report on the results of 9-10 month long campaign (September to June) characterizing the chemical composition of fresh snow sampled at Alert. The first paper presented the data and compared it to simultaneous

measurements of aerosol composition to assess the efficiency of air to snow deposition for the different analytes. Here the focus is application of PMF and the FLEXPART transport modeling tool to assess source regions for the various chemical compounds measured in the snow. This is a solid piece of work, though I feel that the manuscript is less accessible than it could be (more on that below). I also suggest that the authors should consider changing the emphasis in several places in the discussion, to better reflect a lot of other recent (and also pioneering) work on related topics. A very good example of this arises as early as the abstract, where the finding that BC in the high Arctic during winter is dominantly from anthropogenic sources (fossil fuel combustion) and not biomass burning is highlighted. In section 3.2.2 their analysis refines this even more and points to sources in Eurasia for nearly all of this anthropogenic BC. To me, this is basically rediscovering some of the very early findings from a host of "Arctic Haze" investigations initiated in the 1970s which documented that the Haze was largely pollution, it was significantly absorbing due to BC, and much of it came from relatively high latitudes in Europe and Russia. Authors note that their work is focused on snow rather than aerosol, yet they explicitly assert that the snow is providing constraint on aerosol sources, so this "finding" is reassuring but perhaps not so exciting as to merit being the only factor from the PMF to be called out in the abstract. This statement about BC in the abstract notes that it is a "light-absorbing compound critical to the Arctic radiative balance" which is certainly true. However, the AMAP, 2015 assessment (cited frequently in this manuscript) points out that a suite of CTMs all agree that Asian sources dominate the atmospheric burden and climatic impact of BC in the Arctic. Most likely this apparent discrepancy is due to the highly stratified Arctic winter time troposphere, allowing Eurasian BC sources to be dominant in lower levels (sampled at surface aerosol sites and scavenged by mid- to low-level clouds) while Asian BC is at higher altitudes. In any case, I find the present result that essentially no Asian BC gets to Alert within 10 days more interesting than seeing very little biomass burning smoke in the high Arctic during winter. Response: We thank the referee for their time in commenting on this manuscript. We agree that there is existing evidence pointing to

a significant anthropogenic influence on particle black carbon (BC) levels in the Arctic. However, we would like to note that some recent studies specifically of the sources of BC in Arctic snow samples (i.e., Hegg et al., 2009; Hegg et al. 2010) have suggested that biomass burning is the dominant source of BC found in Arctic snow. Furthermore, as per the comment of referee #2, it has been suggested that the dominant source of BC to Arctic snow may vary by location or time of year. Thus, we think that additional evidence on the sources of BC to Arctic, specifically the portion that is deposited to Arctic snow, is important to discuss. We do agree that the manuscript would benefit by expanding the focus beyond BC. Several revisions have been made to the manuscript to give more attention to other chemical species critical to the Arctic atmosphere, as suggested above and in following referee comments. We have also added discussion on the geographic source of BC, with the findings of this paper indicating a largely central Eurasian source as opposed to an East Asian source. We would like to thank the referee for this suggestion.

Referee Comment: A very interesting finding in this work is the lack of a strong anthropogenic sulfate signal. Arctic Haze "comprises a varying mixture of sulfate and particulate matter and, to a lesser extent, ammonium, nitrate, dust, and black carbon (e.g., Li and Barrie, 1993; Quinn et al., 2002)" (Quote from chapter 4 of AMAP, 2006; another work cited several times in this manuscript. This statement is also repeated nearly verbatim on page 2 lines 10-11 of this manuscript.) This may reflect imperfect air-snow transfer of a defining characteristic of the Arctic winter-time troposphere, greatly enhanced sulfate, or possibly strong impact from volcanic sources in this particular year (suggested by the authors, but not very convincingly). Critically assessing air to snow transfer of sulfate would provide a nice link to the first paper in this series. However, the missing Arctic Haze sulfate signal could also reflect problems arising from sampling fresh snow from elevated snow tables (see more on this in first detailed comment below). Response: We agree with the referee that the apportionment of sulphate in this study is interesting. While the majority of sulphate is apportioned to Factor 7, sulphate, a significant mass, 24 $\mu$g/m2/period, is also apportioned to Factor 3, BC.

Compared to the mass apportionment of BC to Factor 3, 1.4 $\mu$g/m2/period, this gives a ratio of about 17 mass/mass SO42-/BC. This appears to be similar to the ratio typically observed in Arctic Haze of 10-20 mass/mass (e.g., Hopper, Worthy, Barrie, and Trivett, 1994; Sharma, Lavoué, Chachier, Barrie, and Gong, 2004; Gong et al., 2010 to name a few). Thus, the SO42- apportioned to Factor 3, BC, seems appropriate for Arctic Haze. Furthermore, SO42- was observed to have significant mass loading of 46 $\mu$g/m2/period on Factor 6, non-crustal metals, also considered to be anthropogenic in origin. As the referee noted, the previous publication Macdonald et al. (2017) found SO42- to show a higher deposition velocity than BC, especially in the warmer fall months. Several factors likely contributed to this trend. A potential explanation could be that heightened SO2 scavenging in the fall lead to an increased level in the snow relative to BC. Specifically sulfate/SO2 from volcanic sources prevalent in the fall may have been scavenged more readily than BC, resulting in an enhanced SO42- deposition velocity and the identification of a separate SO42- dominated factor in the fall. Additional research would be required to confirm this hypothesis (i.e. the SO2 would have to be oxidized to sulfate in the precipitation or snow), but we believe it is a reasonable explanation of the observations of these two papers. The discussion of Factor 7 has been revised to expand on these points and we would like to thank the referee for their suggestion. Please see the response to the first detailed comment for a discussion on the impact of undercatch on the sulphate signal. Gong, S. L., Zhao, T. L., Sharma, S., Toom-Sauntry, D., Lavoué, D., Zhang, X. B., Leaitch, W. R., and Barrie, L. A.: Identification of trends and interannual variability of sulfate and black carbon in the Canadian High Arctic: 1981-2007, J. Geophys. Res.-Atmos., 115 (D07305), 1–9, doi:10.1029/2009JD012943, 2010. Hopper, J. F., Worthy, D. E. J., Barrie, L. A., and Trivett, N. B. A.: Atmospheric observations of aerosol black carbon, carbon dioxide and methane in the high arctic, Atmos. Environ., 28, 3047–3054, doi:10.1016/1352-2310(94)90349-2, 1994. Sharma, S., Lavoué, D., Chachier, H., Barrie, L. A., and Gong, S. L.: Long-term trends of the black carbon concentrations in the Canadian Arctic, J. Geophys. Res.-Atmos., 109 (D15203), 1–10, doi:10.1029/2003JD004331, 2004.

Referee Comment: One final example of a finding that is perhaps misinterpreted or at least somewhat misrepresented is the attribution of PMF factor 2 to local dust. V, Se, and As are generally considered to be dominated by anthropogenic emissions, and in fact the authors point this out in their later discussion of factor 6. In particular, finding V to be enriched in Arctic Haze caused Ken Rahn to reassess, and basically refute (Rahn et al. 1985 in Atmos. Environ., see also AMAP, 2006, chapter 4), his own early suggestion that the haze was mostly dust from Asia (Rahn et al., 1977 in Nature). Mosher et al., 1993 used V to show that emissions from the generators at the DYE 3 radar station probably had a subtle but persistent impact on aerosol measurements made during the DGASP campaign. (Pretty well established that V is a tracer of oil combustion, in fact the authors point this out in discussion of factor 7.) Given the correlation between factor 2 and winds from the main station at Alert, it would seem plausible that local pollution, and not just local dust, is part of this factor. Response: While we agree that V, Se, and As are typical of anthropogenic sources they also occur in dust sources. The ratio of these metals to Al in Factor 2, crustal metals, were 0.0016, 0.0031, and 0.00081 m/m for V, Se, and As, respectively. Soils vary significantly in composition, but typical ratios to Al are 0.0012 - 0.0016, 0.000001 - 0.00027, and 0.00002 m/m for V, Se, and As, respectively (Taylor, 1964; Barrie, den Hartog, and Bottenheim, 1989; Masson-Delmotte et al., 2013). Measurements of local crustal sources in the Arctic have also seen ratios to Al of 0.0013 and 0.00013 m/m for V and As, respectively (Se not measured) (Barrie, den Hartog, and Bottenheim, 1989). As discussed in the manuscript, this gives enhancement ratios of approximately unity for V, 11-5000 for Se (note this large range is a result of the high variability in crustal measurements), and 6-37 for As. Thus, the loading of V in particular on this factor is very reasonable for a crustal source. The loadings of Se and As are higher than for typical soils but given the variability seen across crustal sources both could still be explained by a crustal source. Furthermore, the raw unapportioned concentration measurements of V, Se, and As all correlate to Al with Pearson's correlation coefficients of 0.91 or higher. Timeseries of these analytes are provided in the supplemental. An important distinction in this analysis is that the

V, Se, and As measurements being discussed are the insoluble portions (as noted in the original manuscript page 7 lines 31-32, and revised manuscript page 7 lines 2-3). The soluble portion of these metals was often below detection limits with weak signal-to-noise and therefore was excluded from the apportionment analysis (note that the portion considered as "soluble" would include soluble metals as well as insoluble metals associated with particles capable of passing through a 0.45 $\mu$m filter; Macdonald et al., 2017 provides further details about this analysis). Of these three metals soluble As had the highest number of measurements about detection limit. The soluble As time series correlated best with Factors 3 and 6, black carbon and non-crustal metals. The limited data available for soluble metals contributes a high degree of uncertainty to any discussion of their potential apportionment, but their correlation with these anthropogenic factors may indicate that the anthropogenic sources of these metals were mostly captured in the soluble measurements while the insoluble measurements represent a largely crustal source.

Referee Comment: Regarding comment about accessibility of the manuscript, the very detailed description of PMF in section 2.4.1 and section 3.1 describing how 7 factors were ultimately selected is too lengthy for a journal like ACP, especially considering that the algorithm is publicly available and presumably well described in EPA documents and Norris et al., 2014. Material in the supplemental showing the changes as additional factors are considered is well done, but not distracting to someone reading the paper who may be less interested in statistical details. Response: We agree with the referee that a detailed description of PMF is not required within the manuscript, given the target audience of this paper. Though we do think that this information is vital to be included in all papers with PMF analyses for reproducibility and transparency. Thus, portions of section 2.4.1 and 3.1 have been moved to the supplemental.   Response to Detailed Comments Referenced to Page/Line #(s) in the original manuscript: 2/31-3/4 Referee Comment: The first paragraph of section 2.1 probably needs to be expanded to provide a few additional details about sampling and data screening. In particular, in Macdonald, 2017 the chemical fluxes in January and

[Figure]

February were excluded in all analyses due to indications that the snow tables suffered extreme undercatch during high winds in mid winter. However, in this manuscript these data are retained, the PMF is conducted on "flux per snowfall event" rather than concentration or flux per day, and spikes in several of the factors during January and February were used to support attribution of the factor to source. Authors need to justify this pretty large change in assessment of data quality (or stick with original decision and leave mid winter out of the PMF). As noted above, I wonder if low fluxes due to snow undercatch obscured the expected winter peak in sulfate flux. Response: Additional details on the sampling procedure have been provided in the supplemental, revised section S1. We do not believe the undercatch noted in the previous study detrimentally impacted this sourcing analysis. The composition of the snow throughout January and February is not expected to be impacted by undercatch, simply the total volume of snow. Underestimation of all analytes for a few dates does not greatly impact the apportionment of a PMF analysis, since this analysis focusses primarily on the relative variation in analytes rather than their magnitude. The profiles of the identified factors should be largely unaffected; however, the temporal flux contributions may be underestimated across all factors for the dates of interest. Furthermore, the source regions identified for each factor by weighted FLEXPART analysis may have understated the impact of source regions prevalent on those dates, but the peaks identified outside of this period should not be affected and are still valid episodes. To better understand the impact of using snow flux instead of concentration three PMF analyses were completed: based on snow concentration, flux per period, and flux per day. The results of these auxiliary runs have been provided in this paper, moved to the supplemental per the referees' suggestions. The concentration PMF factor profiles were found to be highly consistent with those of the flux per snowfall analysis considered in the manuscript. The factor compositions agreed with Pearson's correlation coefficients of 0.97 or higher and contributions agreed with correlations of 0.60 or higher. If the uncertain January and February dates were removed the correlation of the factor contributions between the concentration and flux per snowfall

PMF analyses only changed by less than 6%. Furthermore, the primary evidence used in the identification of the PMF factors in this manuscript was composition, which does not appear to have been impacted by the underestimation of flux based on undercatch in January and February. Specifically looking at sulphate, the concentration time series is very similar to that presented for flux, with a Pearson's correlation of 0.76. Both show a very distinct fall peak with small episodic peaks in winter and spring. Neither show the typical Arctic Haze trend with a broad peak throughout the winter, as observed for BC. For reference, the concentration PMF results are provided in the supplemental and a complete record of the measured concentrations provided in Macdonald et al. (2017). Overall, we chose to include these time periods so as to not lose potential information about sources during this important time of the year. A brief note on this topic has been added to the revised manuscript. (revised manuscript page/line(s): 3/15-17) Figure 3 Referee Comment: Figure 3 probably needs to be modified, given its central role in attributing factors to likely sources. All 7 panels share a lot of similarities that tend to draw the eye as, or even more, strongly than small differences pointed out in the text in section 3.2. Probably the biggest problem is the bulleye very close to Alert in all of the panels. This is largely a geometric artifact reflecting that every particle released from the receptor site has to pass through a very small number of cells surrounding that site. I am pretty sure that Stohl and/or Burkhart have recognized this issue and have a recommended weighting scheme that reduces this bias (lower weights for cells closer to release site). Another minor point is that the green triangles and square in the panel for factor 7 are very hard to find (especially the Smoking Hills square). And the label under color bar should be Residence Time (not Residential), and there has to be some huge multiplier on the scale (max is not just 30 seconds) Response: While we agree that the plots in Figure 3 do share some similarities, we do not believe this is reason to change them. We agree that the "bullseye" on Alert is the result of all tracers being initialized at this location; however, it is correct to say that Alert and the surrounding area is a significant potential source/influencing area for all factors. Reducing the weighting on this area may help in identifying long-range

sources but we believe it is important to emphasize that all factors could potentially be strongly influenced by local activities. Also, we find it interesting that some factors seem to show common source/influence areas. Specifically, Factors 3, 5, and 6 all likely have anthropogenic origins and all show similar source regions, with some small exceptions. These source regions show a distinct contrast from those of Factors 1, 2, 4, and 7 which appear to be more dependent on Arctic sources/influences. We agree that the symbols denoting Alert and volcanic sources are quite small (as noted by both referee #1 and 3). This was done so as to not block a significant portion of the trajectory plot. This figure will be uploaded as a high-resolution image allowing readers with difficulty seeing these symbols to simply zoom in as needed, without sacrificing the details of the trajectory plot. The legend has been corrected to residence time. The scale has been converted to a unitless relative residence time since interpretation of the actual residence time requires information on the cell size. (revised page 10) 1/23 Referee Comment: AMAP 2011 was updated in 2017, probably should cite that report Response: We thank the referee for this note. The reference to AMAP 2011 has been updated to the 2017 revision and this revision reviewed for any changes in relevant sections. 2/6-8 Referee Comment: Not sure how the concluding phrase about snow as a critical reservoir logically follows the first part of this sentence. Original Line: Particles entering the Arctic atmosphere can be removed only by atmospheric transport or deposition, and the deposition processes are much slower in the winter than in the summer; thus Arctic snow is a critical reservoir within the Arctic system. Response: We agree that this line was poorly phrased. The line has been revised to clarify as follows: Particles entering the Arctic atmosphere in winter can be removed only by atmospheric transport or deposition in snow where they can be retained for an extended time; thus Arctic snow is a potentially critical reservoir within the Arctic system. (2/17-19) 2/8-17 Referee Comment: Given the vast literature on Arctic Haze, it is unclear how the references in this section were selected. Personally, I would like to see some of the very early work cited. At a minimum, indicate that AMAP, 2006 is a review paper and readers should see references cited therein. Response: We

agree that additional sources should be included, but recognize that this is not meant to be comprehensive review paper. The following references have been added to text; furthermore, we have urged the reader to see the references within existing review papers for further information. Barrie, L. A.: Arctic air pollution: An overview of current knowledge, Atmos. Environ., 20 (4), 643–663, doi:10.1016/0004-6981(86)90180-0, 1986. Mitchell, J. M.: Visual range in the polar regions with particular reference to the Alaskan Arctic, J. Atmos. Terr. Phys., 17, 195–211, 1957. Rahn, K.A., Borys, R., and Shaw, G. E.: The Asian source of Arctic Haze bands, Nature, 268, 713–715, doi:10.1038/268713a0, 1977. Shaw, G., and Wendler, G.: Atmospheric turbidity measurements at McCall Glacier in northern Alaska, B. Am. Meteorol. Soc., 53 (5), 510, 1972. 3/4 Referee Comment: The last phrase after the comma is very much a matter of personal opinion. I suggest ending sentence with a period after flux (see first detailed comment above). Original Line: The use of a snow table allowed the deposition area associated with each sample to be recorded and used in the con-version of measured concentration to flux, which provided a considerable advantage over previous snow sampling campaigns. Response: This line has been revised per the referee's suggestion. (3/13-15) 3/20-21 Referee Comment: Reword this to make argument more clear, and possibly consider different wording for "under-exaggerate". Are you saying that you tossed BDL samples to make the S/N higher than it probably should have been? Original Line: The signal-to-noise (S/N) of each analyte was also calculated to indicate the strength of each measurement. Given the enhanced uncertainty of below MDL and missing values, these data points were excluded so as to not under-exaggerate the S/N (Norris et al., 2014). Response: The calculation for signal-to-noise was adopted from the EPA PMF guide (Norris et al., 2014 equation 5-3 and 5-4) and is suggested for environmental data. This approach is meant to recognize that environmental data often include some missing or even negative values which, with the older PMF4 S/N calculation, would have artificially decreased the S/N ratio. This line was revised to clarify. (4/3-6) 5/32-6/6 Referee Comment: Is this needed? Results from PCA are not shown, and appear to be mentioned in passing

just once more in the manuscript (page 8, line 8) Response: The paragraph mentioned provides a description of the principal component analysis and how it was applied to this data. As the referee notes, the results of this analysis are only provided in the supplemental and are only briefly discussed in the text. Per the referee's suggestion the bulk of this paragraph has been moved to the supplemental, section S4.3. 6/19 Referee Comment: residential-→residence Response: Editorial comment addressed in revised text. (6/3) 9/1 Referee Comment: Enhancement of Mg above the SS ratio by a factor of 1.6 is a big difference that would suggest an additional Mg source. Same is true for SO4, but excess is expected. Response: The enrichment of Mg2+ and SO42- has been noted in the text. The enrichment of Mg2+ was found to be consistent even for PMF analyses with a greater number of factors which does not suggest a missing factor is responsible for the enrichment. Furthermore, similar enrichment of Mg2+ in a sea salt factor was also observed by Krnavek et al. (2012). The uncertainty of these enrichment ratios has been included in the text, presented as the PMF 25th and 7th bootstrapping results. (11/11-18) Krnavek, L., Simpson, W. R., Carlson, D., Domine, F., Douglas, T. A., and Sturm, M.: The chemical composition of surface snow in the Arctic: Examining marine, terrestrial, and atmospheric influences, Atmos. Environ., 50, 349–359, doi:10.1016/j.atmosenv.2011.11.033, 2012. 9/14-15 Referee Comment: The residence time plot suggests that the middle of the GrIS is a stronger source for this factor than Norwegian Sea or North Atlantic, probably partly due to geometric artifact mentioned earlier. Response: It has been noted in the text that the influence of the area immediately around Alert may be over-exaggerated in Figure 3. While it is true that the Greenland ice sheet is a potential area of influence for Factor 1, the ice-free Norwegian sea and Northern Atlantic ocean are also potential areas of influence and we believe are a more probable potential source region. 10/Figure 1 Referee Comment: Please explain what the bars on this plot are showing more clearly. What is the time component indicated by "/period"? Original Line: Factor profiles. Error bars show the 25th and 75th percentiles of the bootstrapping analysis. Flux contributions below 0.00001 $\mu$g/m2/period are not shown. Response: The percentile

and mass loading to each factor is the typical method of describing PMF results. A thorough discussion of how to interpret these results is provided in the EPA PMF guide (Norris et al., 2014). However, the author recognizes that not all readers will be familiar with such analyses. For clarity, the Figure 1 caption has been revised as follows: Factor profiles. The loading of each analyte to each factor is provided as the portion of their flux apportioned to that factor as well as the percentage of the analyte's total flux (mass/mass) apportioned to that factor. Error bars on the percentage loading show the 25th and 75th percentiles of the bootstrapping analysis. Flux contributions below 0.00001 $\mu$g/m2/period are not shown. Metals with a charge are those measured by IC, others are insoluble portions measured by ICP-MS. (revised page 8) Section 3.1 paragraph one describes the flux per snowfall period metric used. 12/Figure 3 Referee Comment: Why not label the panels by source name rather than factor #? Response: Figure 3 has been updated to include full factor names. (revised page 10) 13/29 Referee Comment: There have been a lot of papers on emissions from fires (lab, prescribed, and wild) since 2009. Liu et al., 2017 in JGR maybe most recent. This one does not include BC, but provides access to many of the papers between 2009 and 2017. Response: This section has been revised to include references to the following more recent studies: (13/31-32) Liu, X., Huey, L. G., Yokelson, R. J., Selimovic, V., Simpson, I. J., Müller, M., Jimenez, J. L., et a;.: Airborne measurements of western U.S. wildfire emissions: Comparison with prescribed burning and air quality implications, J. Geophys. Res. Atmos., 122, 6108–6129, doi:10.1002/2016JD026315, 2017. May, A. A., McMeeking, G. R., Lee. T., Taylor, J. W., Craven, J. S., Burling, I., Sullivan, A. P., et al.: Aerosol emissions from prescribed fires in the United States:A synthesis of laboratory and aircraftmeasurements, J. Geophys. Res. Atmos.,119,11,826–11,849, doi:10.1002/2014JD021848, 2014. 14/1-16 Referee Comment: Hirdman et al. 2010 (2 papers, in ACP) and Stohl et al 2006 (JGR) have shown similar. They probably should be cited. Response: The following references have been added to Section 3.2.3: (15/3) Hirdman, D., Burkhart, J. F., Sodemann, H., Eckhardt, S., Jefferson, A., Quinn, P. K., Sharma, S., Ström, J.,

and Stohl, A.: Long-term trends of black carbon and sulphate aerosol in the Arctic: Changes in atmospheric transport and source region emissions, Atmos. Chem. Phys., 10, 9351–9368, doi:10.5194/acp-10-9351-2010, 2010. Stohl, A., Berg, T., Burkhart, J. F., Fjæraa, A. M., Forster, C., Herber, A., Hov, Ø., et al.: Arctic smoke – record high air pollution levels in the European Arctic due to agricultural fires in Eastern Europe, Atmos. Chem. Phys., 7, 511–534, doi.org/10.5194/acp-7-511-2007, 2007. 16/1 Referee Comment: delete "both" Response: Editorial comment addressed in revised text. (15/20) 16/3 Referee Comment: delete "to" Response: Editorial comment addressed in revised text. (15/22) 16/3-4 Referee Comment: There have been a lot of papers on emissions from fires (lab, prescribed, and wild) since 2009. Liu et al., 2017 in JGR maybe most recent. This one does not include BC, but provides access to many of the papers between 2009 and 2017. Response: See response to detailed comment 13/29 above. 17/11 Referee Comment: Why not say "via N2O5 hydrolysis in the aerosol phase" instead of "NO3-radical chemistry"? Original Line: The mid-winter peak in this factor may be linked to NO3- formation via NO3-radical chemistry, which is considered to dominate Arctic NO3- chemistry during the night (Morin et al., 2008). Response: This line was revised as suggested. (17/1-2) 18/14 Referee Comment: Laing et al. 2014 is not original source of this fact, Rahn probably closer, but maybe even he used someone else's earlier work Original Line: Non-crustal Se is typically considered to be a tracer of coal combustion and V a tracer of oil combustion (Laing et al., 2014). Response: We agree that the original reference should be provided. The following references have been added, which we believe to be some of the earliest to discuss this topic. (18/11-12, 20/1-2) Key, C. W., and Hoggan, G. D.: Determination of trace elements in fuel oils, Anal. Chem., 25 (11), 1673–1676, doi:10.1021/ac60083a027, 1953. Rahn, K. A.: Sources of trace elements in aerosols – An Approach to clean air, Ph.D. thesis, University of Michigan, 1971. 18/20-21 Referee Comment: Fact that FLEXPART rarely reaches any of these volcanoes is a little problematic. Response: The Factor 7, Sulphate, section has been revised to address several comments from all referees. We recognize that Figure 3 does not show high influence from

the noted volcanic sources for Factor 7; however, this plot only represents a ten-day back trajectory and does seem to indicate that Factor 7 is more likely a dominated by relatively local sources rather than long-range anthropogenic sources. Furthermore, these plots only highlight areas over which the trajectories passed within 500 m of the surface (as noted in section 2.4.2). This approach is useful for identifying ground-level sources which could have reasonably impacted the air mass. However, volcanic sources can impact air masses to a much great height, given the heat and velocity of the emitted plume; thus, trajectories at a greater height should be considered. We have reviewed the FLEXPART influence plot for Factor 7 for trajectories within 10 km of the surface and this plot does show greater potential influence from the BárÃřarbunga volcano in Iceland and the Smoking Hills in Canada. (section 3.2.7) 21/7 Referee Comment: seasonally-→seasonal Response: Editorial comment addressed in revised text. (22/22)

Please also note the supplement to this comment:
https://www.atmos-chem-phys-discuss.net/acp-2017-718/acp-2017-718-AC1-supplement.pdf

---

## Author Comment (AC2) · 16 Dec 2017

Temporally-Delineated Sources of Major Chemical Species in High Arctic Snow – Response to Anonymous Referee #2

Referee comments received and published: 25 September 2017 (quoted below)

We would like to thank Referee #2 for their detailed comments and discussion. We greatly appreciate the care with which the referee has reviewed this manuscript and the improvements gained through their insight. Response to Referee Discussion Referee Comment: Macdonald et al describe the results of positive matrix factorization of snow chemical composition measurement data from Alert, Nunavut in order to determine the prominent sources influencing the snow composition. Given changing Arctic source

[Figure]

emissions with sea ice loss and increasing development, this is an important topic. A thorough description of the data analysis is provided. My main concerns, described below, surround the discussion of the results. The main result highlighted in the abstract and conclusions is that the BC is primarily from fossil fuel burning, rather than biomass burning influence. This is not surprising since the study focuses on snow samples collected from Sept. 14, 2015 to Jun. 1, 2015, outside of the main summertime wildfire period. In several places in the paper (last paragraph of Section 3.2.3, part of Sec. 3.3, and P21 L 11-14), it is stated that these results "disagree" with previous snow chemical composition measurements that showed greater biomass burning influence, proving "contradicting snow BC apportionment findings". The authors do note the influence of seasonality and changes in annual wildfire frequency and severity on contributions of biomass burning BC. However, because the references that the authors are comparing to correspond to different times and locations, a simple comparison of the percentages of biomass burning vs fossil fuel influence is not appropriate (e.g. Table 2), without an in-depth analysis of fire locations, frequency, and timing, as well as air mass trajectories associated with the various sampling sites. I would expect that the contribution of biomass burning vs fossil fuel likely depends on the site, season, and year. Therefore, I suggest revising the discussions and comparisons to provide these results as another study that points to the variability in BC source contributions, rather than suggesting that they "disagree with" or "contradict" previous results, which gives the idea of invalidating previous work, which instead may simply be different due to different timing and location. As part of this revision of the discussion, I suggest removing Table 2, or if the authors feel strongly about keeping this comparison, then information about timing, location(s), and wildfire influence (from fire maps and air mass trajectory analysis, presumably, or statements from previous papers) should be included. In addition, a more thorough literature search is needed if the authors mean for this to be a comprehensive comparison. Response: We agree with the referee that the discussion and tone of the listed sections should be changed. While we did try to keep our literature comparison to mostly studies of similar seasons and locations, we do agree that the sources of

BC appear to be dependent on several factors. The paper has been revised to avoid statements that these results contradict those of previous studies and we instead state that they highlight the importance of understanding the variability of BC sources to Arctic snow. In general, the focus of the paper has been shifted away from BC, per the comments of the referees. Furthermore, Table 2 of the refereed document has been removed. This table was meant to be illustrative rather than comprehensive, but we agree that it is not needed.

Referee Comment: This is a complementary paper to the recent Macdonald et al (2017) ACP manuscript that describes the deposition of the same chemical species to the snowpack, with snow mixing ratios and fluxes of these species described. In that paper, Figure 1 shows time series over the same period of Sept 2014 to Jun 2015 for the following "key analytes" (as described in that paper), grouped according to time series correlations: Black carbon, methanesulfonate, $C_2O_4^{2-}$ & $NH_4^+$, sea salt, NSS-sulfate, nitrate, NSS-$K^+$ & NSS-$Br^-$, and crustal metals; this is quite similar to the time series of the 7 factors (salt, dust, BC, carboxylic acids, nitrate, metals, and sulfate) in Figure 2 of the current paper. Despite this overlap, little discussion was included in the previous manuscript regarding likely sources. Response: This manuscript is meant to be a companion to the previous paper (Macdonald et al., 2017) mentioned by the referee. The first paper outlines the measurements and analysis in greater detail and provides a comparison with concurrent atmospheric measurements. This paper expands on the previous, focussing on sources of these analytes to Arctic snow. Per the suggestion of the referees some additional references to the first paper and over-arching discussion have been added to the revised manuscript. The time series provided in Macdonald et al. (2017) are grouped into related species or those with similar measured ranges, to facilitate plotting. All apportioned time series are also provided in this paper's supplemental. Macdonald, K. M., Sharma, S., Toom, D., Chivulescu, A., Hanna, S., Bertram, A. K., Platt, A., Elsasser, M., Huang, L., Tarasick, D., Chellman, N., McConnel, J., Bozem, H., Kunkel, D., Ying Duan, L., Evans, G. J., and Abbatt, J. P. D.: Observations of atmospheric chemical deposition to high Arctic snow, Atmos.

Chem. Phys., doi:10.5194/acp-17-5775-2017, 2017.

Referee Comment: The authors are encouraged to do a more thorough literature search for previous Alert snow, aerosol, and trace gas studies that likely will support their source apportionment findings and provide evidence for greater certainty for source identification. Some appropriate papers (not meant to be comprehensive) are noted below for discussion of specific factors. While not temporally resolved, Krnavek et al 2012 (Atmos. Environ.) provide a detailed source apportionment of marine, terrestrial, and atmospheric influences on Arctic surface snow composition. Most notably, the authors do not cite or compare to Toom-Sauntry and Barrie (2002, Atmos. Environ) who previously collected weekly snow samples at Alert from 1990 to 1994 and measured inorganic and organic ions; this paper is highly relevant to the current work! Response: The study by Toom-Sauntry and Barrie (2002) is referenced in the previous paper discussing these snow measurements (Macdonald et al., 2017). A comparison of the snow measurements from this campaign to those in previous studies, including Toom-Sauntry and Barrie 2002, is included in Macdonald et al. (2017) supplemental section S1. The trends and absolute values of major ions measured in snow in this study were mostly found to be consistent with those observed by Toom-Sauntry and Barrie, 2002. However, we agree that further discussion of how these measurements compare to those of Toom-Sauntry and Barrie within this paper is also warranted. Sections 3.2.1 and 3.2.7 have been revised to include this discussion. We thank the referee for suggesting Krnavek et al. (2012). We have reviewed this paper and incorporated it into our discussion. We have also expanded our literature review of other related studies. The following references have been added to the manuscript: Barrett, T. E., Robinson, E. M. Usenko, S. and Sheesley, R. J.: Source contributions to wintertime elemental and organic carbon in the western Arctic based on radiocarbon and tracer apportionment, Environ. Sci. Technol., 49 (19), 11,631–11,639, doi:10.1021/acs.est.5b03081, 2015. Breider, T. J., Mickley, L. J., Jacob, D. J., Wang, Q., Fisher, J. A., Chang, R. Y.-W., and Alexander, B.: Annual distri-butions and sources of Arctic aerosol components, aerosol optical depth, and aerosol

absorption, J. Geophys. Res.-Atmos., 119, 4107–4124, doi:10.1002/2013JD020996, 2014. Doherty, S. J., Warren, S. G., Grenfell, T. C., Clarke, a. D., and Brandt, R. E.: Light-absorbing impurities in Arctic snow, Atmos. Chem. Phys., 10, 11,647–11,680, doi:10.5194/acp-10-11647-2010, 2010. Dou, T., Xiao, C., Shindell, D. T., Liu, J., Eleftheriadis, K., Ming, J., and Qin, D.: The distribution of snow black carbon observed in the Arctic and compared to the GISS-PUCCINI model, Atmos. Chem. Phys., 12, 7,995–8,007, doi:10.5194/acp-12-7995-2012, 2012. Hirdman, D., Burkhart, J. F., Sodemann, H., Eckhardt, S., Jefferson, A., Quinn, P. K., Sharma, S., Ström, J., and Stohl, A.: Long-term trends of black carbon and sulphate aerosol in the Arctic: Changes in atmospheric transport and source region emissions, Atmos. Chem. Phys., 10, 9351–9368, doi:10.5194/acp-10-9351-2010, 2010. Krnavek, L., Simpson, W. R., Carlson, D., Domine, F., Douglas, T. A., and Sturm, M.: The chemical composition of surface snow in the Arctic: Examining marine, terrestrial, and atmospheric influences, Atmos. Environ., 50, 349–359, doi:10.1016/j.atmosenv.2011.11.033, 2012. Law, K. S., Stohl, A., Quinn, P. K., Brock, C. A., Burkhart, J. F., Paris, J.-D., Ancellet, G., et al.: Arctic air pollution: New insights from POLARCAT-IPY, B. Am. Meteorol. Soc., 95 (1), 1873 − 1895, doi:10.1007/BF00138862, 2014. McConnell, J. R., Edwards, R., Kok, G. L., Flanner, M. G., Zender, C. S., Saltzman, E. S., Banta, J. R., et al.: 20th-Century industrial black carbon emissions altered Arctic climate forcing, Science, 317, 1381–1384, doi:10.1126/science.1144856, 2007. Pratt, K. A., Custard, K. D., Shepson, P. B., Douglas, T. A., Pöhler, D., General, S., Zielcke, J., et al.: Photochemical production of molecular bromine in Arctic surface snowpacks, Nat. Geosci., 6 (5), 351–356, doi:10.1038/ngeo1779, 2013. Sharma, S., Ishizawa, M., Chan, D., Lavoué, D., Andrews, E., Eleftheriadis, K., and Maksyutov, S.: 16-year simulation of arctic black carbon: Transport, source contribution, and sensitivity analysis on deposition, J. Geophys. Res.-Atmos., 118, 943–964, doi:10.1029/2012JD017774, 2013. Stohl, A., Berg, T., Burkhart, J. F., Fjæraa, A. M., Forster, C., Herber, A., Hov, Ø., et al.: Arctic smoke – record high air pollution levels in the European Arctic due to agricultural fires in Eastern Europe, Atmos. Chem. Phys., 7, 511–534, doi.org/10.5194/acp-7-511-2007, 2007.

[Figure]

Toom-Sauntry, D. and Barrie, L. A.: Chemical composition of snowfall in the high Arctic: 1990–1994, Atmos. Environ., 36, 2683–2693, doi:10.1016/S1352-2310(02)00115-2, 2002. VanCuren, R. A., Cahill, T., Burkhart, J., Barnes, D., Zhao, Y., Perry, K., Cliff, S., and McConnell, J. R.: Aerosols and their sources at Summit Greenland - First results of continuous size- and time-resolved sampling, Atmos. Environ., 52, 82–97, doi:10.1016/j.atmosenv.2011.10.047, 2012   Response to Detailed Comments – Major Comments Referenced to Page/Line #(s) in the original manuscript: The first paragraph Abstract Referee Comment: Currently, only two results are noted here – the names of the source factors and the fossil fuel source of the BC. Can additional results associated with other factors be mentioned here to highlight this work? Also, please be consistent between the factor names here and throughout the text (e.g. this says "regional dust", but later it is discussed that the dust is likely local). Response: Per the referees' suggestion, the abstract has been revised to briefly summarize all factors resolved rather than focussing on Factor 3, BC. We agree that factor naming should be consistent throughout. The revised manuscript uses the following names when referring to Factors 1 to 7, respectively: sea salt, crustal metals, black carbon, carboxylic acids, nitrate, non-crustal metals, and sulphate. Section 3.2.1 – Factor 1 (Marine Sea Salt) Referee Comment: Is there seasonal dependence to the Br-enrichment factor? There is well-known multiphase bromine chemistry that occurs in the Arctic in the spring (see Simpson et al. 2007, ACP). Hara et al (2002, J. Geophys. Res.) conducted a detailed examination of Br- enrichments in Arctic aerosols and may be useful to consider for this work. A neutralization ratio of 0.8 is stated as neutral; what is the uncertainty associated with the calculated ratio? The discussion of the potential sea salt sources is muddled with respect to local vs far away sources and should be clarified, with improved flow in discussing the possibilities. Note that recent work has suggested that aerosols are not produced from frost flowers (Yang et al 2017, ACP; Roscoe et al 2011, J. Geophys. Res.). Were there are open leads upwind of the field site, such that open water was closer to the site? May et al. (2016, J. Geophys. Res.) pointed to sea salt production from leads in

the fall-spring. Response: Br-enrichment is observed in the spring. This observation was discussed in the previous companion paper: Macdonald et al., 2017. The time series of Br- is provided in the supplemental, showing a broad spring peak, and mentioned in the manuscript in section 3.2.5. This peak is not well-predicted by the PMF results. Section 3.2.1 of the revised manuscript has been updated to include a brief mention of Br-enrichment. The neutralization ratio of each factor is summarized in the revised Table 2. This table also includes the ratio calculated from the 25th and 75th bootstrapping. Factor 1 has a neutralization ratio of 0.79 with bootstrapping of 0.75 to 0.84. Section 3.2.1 discussion has been revised to improve flow and clarity. The correlation between Factor 1, sea salt, and local wind speeds was weak, a Pearson's correlation of 0.28. We agree that for local wind speeds to be relevant there must be a local source of sea salt. This could include any local open water, blowing saline snow, or frost flowers; however, we would require more data to confirm the existence of any of these sources at the specified time. The possibility of a frost flower source has been noted as quite uncertain in the revised text. Upon further consideration, we have noted that Factor 1, sea salt, in fact has a stronger correlation with collection period length (Pearson's correlation coefficient of 0.47). The January peak of this factor was one of the longer collection period of the campaign. This may suggest that the deposition of sea salt aerosol was relatively continuous over time; thus, longer collection periods were associated with higher sea salt signatures. However, it should be noted that both of these correlations are fairly weak, so these inferences should be considered uncertain. The 0.28 correlation between Factor 1 and wind speeds has been deemed too weak to include in the revised manuscript (a minimum of 0.3 has been imposed on the values included). Section 3.2.3 – Factor 3 (BC) Referee Comment: The authors should consider the work of Doherty et al (2010, ACP), who measured light-absorbing impurities in âĹij1200 snow samples across the Arctic. Dou et al (2012, ACP) previously compared measured snow BC to simulations of the spatial distribution of snow BC using the GISS-PUCCINI model. Recently, Barrett et al (2015, Environ. Sci. Technol.) used radiocarbon tracers to determine elemental

carbon source apportionment between modern and fossil fuel carbon at Barrow, AK; perhaps some discussion in that work may be helpful here. Response: The suggested references have been added to section 3.2.3. Section 3.2.4 – Factor 4 (Carboxylic Acids) Referee Comment: In the authors' consideration of carboxylic acid sources, they should consult the work of Narukawa et al (2002, Atmos. Environ.) who measured aerosol and surface snowpack dicarboxylic acids at Alert in Feb and April-May 2000. Dibb and Arsenault (2002, Atmos. Environ.) examine snow as a source of acetic and formic acids. Response: Narukawa, Kawamura, and Bottenheim (2002) explored dicarboxylic acid measurements in Arctic aerosol and surface snowpack. Given that this campaign did not include measurements of formate and/or acetate which are the dominant components of Factor 4, we have decided not to include it in the discussion. However, we thank the referee for his suggestion. The Dibb and Arsenault (2002) paper mentioned is already included in this discussion. (16/1-2) Dibb, J. E. and Arsenault, M.: Shouldn't snowpacks be sources of monocarboxylic acids?, Atmos. Environ., 36, 2513–2522, doi:10.1016/S1352-2310(02)00131-0, 2002. Section 3.2.5 – Factor 5 (Nitrate) Referee Comment: The authors cite Morin et al (2008) and Fibiger et al (2016) for nitrate cycling associated with the snowpack. However, Fibiger et al (2016) is a study at Summit, Greenland. There are other appropriate studies at Alert that should be considered in the context of the current work – for example, Ianniello et al (2002, Atmos. Environ) and Beine et al (2002, Atmos. Environ.). Response: The suggested references have been added to the manuscript: Beine, H. J., Honrath, R. E., Domine, F., and Simpson, W. R.: NOx during background and ozone depletion periods at Alert: Fluxes above the snow surface, J. Geophys. Res., 107 (D21), 7-1–7-12, doi:10.1029/2002JD002082, 2002. Ianniello, A., Beine, H. J., Sparapani, R., Di Bari, F., Allegrini, I., and Fuentes, J. D.: Denuder measurements of gas and aerosol species above Arctic snow surfaces at Alert 2000, Atmos. Environ., 36 (34), 5,299–5,309, doi:10.1016/S1352-2310(02)00646-5, 2002. Section 3.2.7 – Factor 7 (Sulfate) Referee Comment: In considering the main sources of snow sulfate, the authors should consult the work of Norman et al (1999, J. Geophys. Res.) who used sulfur isotopes to

determine seasonal aerosol sulfate sources at Alert from July 1993 to Sept. 1994. The authors note that several volcanoes were active over the 2014-2015 season. This factor peaks in the early fall; does this coincide with the volcano activity and associated air mass trajectories (FLEXPART analysis)? Reorganize this section so that there is a clear flow of discussion – currently the authors go back and forth between multiple potential sources. For example, L20-21 and 27-29 seem to be somewhat contradictory as written. L6-7 on P19 seems to be tacked on and should be integrated. Response: Sirois and Barrie (1999), the companion paper to Norman et al. (1999) provides further analysis of aerosol sources. This study is cited within the manuscript. Section 3.2.7 has been revised to improve flow and clarity. The text does state that BárÃřarbunga, a volcano in Iceland, was active during the observed fall peak. The revised section gives details on how this compares with the FLEXPART analysis. Lines 20-21 and 27-29 of the original manuscript have been removed in the revision. Section 3.3 Referee Comment: This section is labeled as "Overall Apportionment", but it is really primarily a discussion of how BC is apportioned between the factors. It may be useful to rename the title of this section, or reorganize and revise the section to make it more evenly about all of the factors. I would suggest a paragraph break at L21, with some reorganization between the two paragraphs. The authors point to mixing state of the particles potentially being important (L23-25), and this could be strengthened by citing previous Arctic studies (e.g. Weinbrunch et al 2012, Atmos. Environ.). Response: Section 3.3 has been heavily revised to include greater discussion of all factors and reduce focus on Factor 3, BC. The apportionment of all analytes has been summarized in the revised Table 3. Figure 4 has also been expanded to show the apportionment of BC, SO42-, and insoluble V. Table 3 Referee Comment: It would be useful to integrate these results into the prior factor discussions (section 3.2). Response: Per the referee's suggestion Table 3 has been moved to revised section 3.2. (now Table 2). Conclusions Referee Comment: The conclusions are very general, with limited discussion of any factor or analyte other than BC. There is an opportunity here to discuss other factors and analytes, particularly with respect to how they may

change in the future, or with respect to uncertainties that should be examined in future work. Response: The conclusions have been revised to discuss other factors and analytes in greater detail.   Response to Detailed Comments – Minor Comments and Technical Corrections Referenced to Page/Line #(s) in the original manuscript: 1/19, 8/13, and other locations Referee Comment: Please clarify text to describe the units used for calculating the percentage. I assume for BC that you are calculating the % based on mass conc? For Na+, for example, are you reporting the fraction of Na+ measured in the snow that was apportioned to the first factor? This isn't currently clear and could be worried more clearly throughout the manuscript where percentages are used. Response: The referee is referring to the percentile loadings of various analytes onto each PMF factor. This represents the portion of total analyte mass apportioned to a single factor. To clarify, "mass/mass" has been added where appropriate, and the first use has been described as follows: The first factor was characterized by high loadings (>75% of total flux mass apportioned to Factor 1) of Na+ and Cl- and 30-45% loadings of Br-, K+, and Mg2+ (Figure 1; Table 2). (revised manuscript page/line(s): 11/8-10) 1/19 Referee Comment: Fix phrasing/sentence structure as snow is not a light-absorbing compound. Original Line: The majority (73%) of the black carbon in snow, a light-absorbing compound critical to the Arctic radiative balance, was found to be the product of fossil fuel burning with limited biomass burning influence. Response: Per the comment above, the abstract has been revised to provide further details about all factors, with less focus on BC. The line above has been removed from the revised abstract. 2/7-10 & 14-15, 15/15, and 17/8-9 Referee Comment: Provide references. Original Lines: 2/7-10: Particles entering the Arctic atmosphere can be removed only by atmospheric transport or deposition, and the deposition processes are much slower in the winter than in the summer; thus Arctic snow is a critical reservoir within the Arctic system. Given the seasonal variability in Arctic aerosol inputs and outputs, a period of enhanced accumulation is typically experienced during the Arctic winter and early spring termed "Arctic Haze". 2/14-15: However, direct measurements of pollutants in Arctic snow have been less common, particularly sampling campaigns of

fresh snow which are less prone to the ambiguities introduced by snowpack collection. 15/15: Possible contributors hypothesized in other studies of arctic carboxylic acids are discussed below including biomass burning, atmospheric or snow photochemical processing, and ocean microlayer emissions. 17/8-9: February to June, 2015, was also characterized by a "bromide explosion", observed as a broad peak in snow and atmospheric Br-. Response: The following reference has been noted to the revised manuscript for Line 2/7-10 (revised page/line: 2/19) AMAP: Acidifying pollutants, Arctic haze, and acidification in the Arctic, Arctic Monitoring and Assessment Programme, Oslo, Norway, 2006. Line 2/14-15 has been revised as follows: "However, direct measurements of pollutants in Arctic snow have been less common, particularly sampling campaigns of fresh snow." per the comment below. (2/26-27) Line 15/15 is simply listing topics that will be discussed in the following section. The following references are provided in the following discussion: Jaffrezo, J.-L., Davidson, C. I., Kuhns, H. D., Bergin, M. H., Hillamo, R., Maenhaut, W., Kahl, J. W., and Harris, J. M.: Biomass burning signatures in the atmosphere of central Greenland, J. Geophys. Res., 103, (D23), 31067-3108, doi:10.1029/98JD02241, 1998. Legrand, M., and de Angelis, M.: Origins and variations of light carboxylic acids in polar precipitation, J. Geophys. Res., 100 (Di), 1445–1462, doi:10.1029/94jd02614, 1995. The following reference has been noted to the revised manuscript for Line 17/8-9 (16/32-33) Macdonald, K. M., Sharma, S., Toom, D., Chivulescu, A., Hanna, S., Bertram, A. K., Platt, A., Elsasser, M., Huang, L., Tarasick, D., Chellman, N., McConnel, J., Bozem, H., Kunkel, D., Ying Duan, L., Evans, G. J., and Abbatt, J. P. D.: Observations of atmospheric chemical deposition to high Arctic snow, Atmos. Chem. Phys., doi:10.5194/acp-17-5775-2017, 2017. 2/15 Referee Comment: Please clarify the phrase "less prone to the ambiguities introduced by snow-pack collection". Response: A comparison of fresh and aged snow sampling was discussed in the previous paper. However, we agree that this line should not be included here without additional clarification or references. In the interest of space, this line has been removed from the revised manuscript. 2/27-28 Referee Comment: Mention measurements data here – otherwise it sounds like the study

includes only PMF and air mass modeling. Original Line: In this context, this paper analyses the sources of chemical components in freshly-fallen snow samples collected over a complete fall-winter-spring at a high Arctic location (Alert, Nunavut), using a combination of Positive Matrix Factorization diagnostics and Lagrangian dispersion modelling. Response: The line has been revised as suggested: "In this context, this paper analyses the sources of chemical components in freshly-fallen snow samples collected over a complete fall-winter-spring at a high Arctic location (Alert, Nunavut) and analysed for a broad suite of analytes, using a combination of Positive Matrix Factorization diagnostics and Lagrangian dispersion modelling." (3/4-7) 7/3 Referee Comment: Is this supposed to be 59 samples (based on P3 L2)? Response: The referee is correct that a total of 59 sets of samples were analysed in the course of this study; however, some collection periods did not provide sufficient snow volume to perform the complete suite of analyses (see referenced Macdonald et al., 2017 for the complete list of sampling dates, completed analyses, and results). Section 2.4.1 of the manuscript explains that the PMF analysis was limited to collection periods with the majority of analytes of interest measured (original manuscript 5/21; revised manuscript 5/12). 7/4 Referee Comment: Clarify wording that you are discussing analyte concentrations and fluxes. Original Line: Three metrics were considered as the basis for this analysis: snow concentration, flux per day, and flux per snowfall (i.e., assuming each sample represented a single snowfall event regardless of the time period over which it occurred, which is known to be true for the majority of samples based on Alert station operator records). Response: The line has been revised to clarify as follows: Three metrics were considered as the basis for this analysis: analyte concentration, flux per day, and flux per snowfall (i.e., assuming each sample represented a single snowfall event regardless of the time period over which it occurred, which is known to be true for the majority of samples based on Alert station operator records). Per the referees' suggestions, details on the PMF analysis have been largely moved to the supplemental, including this line. (revised supplemental page 10) 7/13-14 Referee Comment: This discussion is not intuitive and could be

clarified further. Can we learn about processes from these differences? Original Line: The source contributions identified by the flux per snowfall period analysis were the most readily interpreted as physically realistic factors. Moreover, this metric showed the largest correlation between BC snow and atmospheric measurements (Pearson's correlation coefficients of 0.4, 0.3, and 0.5 for BC concentration, flux per day, and flux per snowfall period, respectively), implying that the flux per snowfall may in general be more closely related to the change in analyte sources over time while concentration and flux per day may be more intrinsically dependent on changes in deposition processes. Response: Per the referee comments, discussion of the concentration and flux/day PMF analyses has been moved from the manuscript to the supplemental. The following has been added to this discussion in the supplemental to clarify: "For example flux per snowfall is likely related to a specific synoptic event, arising from a common location. This will be more useful than concentration given that this value will be affected by the amount of precipitation, and more useful than flux per day that will be affected by the rapidity of snowfall." (revised supplemental page 10-11) 7/17-33, 8/5-9, and Table 1 Referee Comment: I suggest moving these paragraphs to the methods and supplementary information, as they discuss how the authors decided to use seven factors and do not discuss science. The section is also difficult to follow without in-depth knowledge of the method, and without referring back to the methods section frequently. Similarly, I suggest moving Table 1 to the supplementary information. Response: Per the referee's suggestion, this section has been shortened in the manuscript. Table 1 has been left in the manuscript as it lists the analytes included in the PMF analysis and specifically which were considered strong or weak. We agree that the diagnostic properties are not necessary in the manuscript for the target audience of the ACP; however, we do believe it is important to list the analysis main inputs and describe the overall fit of the predicted results. 8/14-15 Referee Comment: These sentences are redundant. Original Line: These compounds are all typical of sea salt, suggesting a marine origin for Factor 1. The composition of Factor 1 was found to be consistent with that of sea salt (Pytkowicz and Kester, 1971). Response:

[Figure]

The intention of these two lines was to convey that the dominant compounds as well as their relative proportions were both consistent with a marine source. We agree that as written this distinction is not clear and the lines become redundant. The line has been revised as follows: These dominant analytes and their relative proportions are consistent with that of sea salt (Pytkowicz and Kester, 1971), suggesting a marine origin for Factor 1. (11/10-11) 9/1-12 Referee Comment: "Compound(s)" should be "ion(s)" here. Also, what are the uncertainties in the enrichment ratios? (These errors should be stated for all enrichment ratios reported in this manuscript.) Response: The word "compounds" has been removed, and typically replaced with "analytes" as to be general. The uncertainty of enrichment ratios have been described in the text using the 25th and 75th bootstrapping analysis results. (11/8-12/3) Figure 1 Referee Comment: Remove "(point)" and "(bar)" on the y axes, as this is already shown in the legend, and "bar" is a unit of pressure. Response: Addressed in revised manuscript. Figure 1 Caption Referee Comment: Provide further description of how to interpret the figure for improved clarity, particularly for those not familiar with PMF. Original Line: Factor profiles. Error bars show the 25th and 75th percentiles of the bootstrapping analysis. Flux contributions below 0.00001 $\mu$g/m2/period are not shown. Response: The percentile and mass loading to each factor is the typical method of describing PMF results. A thorough discussion of how to interpret these results is provided in the EPA PMF guide (Norris et al., 2014). However, the authors recognize that not all readers will be familiar with such analyses. For clarity, the Figure 1 caption has been revised as follows: Factor profiles. The loading of each analyte to each factor is provided as the portion of their flux apportioned to that factor as well as the percentage of the analyte's total flux (mass/mass) apportioned to that factor. Error bars on the percentage loading show the 25th and 75th percentiles of the bootstrapping analysis. Flux contributions below 0.00001 $\mu$g/m2/period are not shown. (revised page 8) Figure 2 Caption Referee Comment: What are the traces normalized to (themselves, other factors?)? What are the units? This caption is not clear. Response: Normalized factor contribution is the metric provided directly by the EPA PMF analysis and is the typical

method used to discuss these results. A thorough discussion of this metric and its interpretation are provided in the EPA PMF guide (Norris et al., 2014). However, the authors recognize that not all readers will be familiar with such analyses. For clarity, the Figure 2 caption has been revised as follows: Normalized factor contribution. The unitless contributions describe the relative magnitude of each factor over time such that the average contribution of each factor is one. (revised page 9) 11/9 Referee Comment: Provide the calculated ratio in parentheses for context. Original Line: Specifically, the modelled ratio of As/Al was seen to be closer to that of local soils (Barrie, den Hartog, and Bottenheim, 1989) than the global typical composition (Taylor, 1964; Masson-Delmotte et al., 2013) with enrichment ratios of 6 and 37, respectively. Response: This primary focus of this line is to convey the greater similarity of the apportioned factor to local soil as compared to typical global soils. This is exemplified with the enrichment ratios provided. The line has been revised to also provide the ratios as follows: Specifically, the modelled ratio of As/Al (0.00081 m/m) was seen to be closer to that of local soils (0.00013) (Barrie, den Hartog, and Bottenheim, 1989) than the global typical composition (0.00002) (Taylor, 1964; Masson-Delmotte et al., 2013) with enrichment ratios of 6 and 37, respectively (6.3-9.5 and 37-58 25th-75th percentiles per bootstrapping analysis). (12/28-31) 13/32 Referee Comment: The neutralization equation is provided on P9, but it is not clear if the same equation is used for the calculation here and elsewhere in the paper. Response: The updated manuscript provides all neutralization ratios in the revised Table 2. It is clarified that the provided formula is used for all calculations. (9/8-11) Table 3 Referee Comment: While there is a footnote defining "Southern Oceans", I suggest renaming to Atlantic & Pacific Oceans, since "Southern Ocean" is a phrase typically referring to near the Antarctic. Response: We agree that the original naming could be misconstrued. This has been revised as "Open Ocean". (revised page 11) Figure 4 Referee Comment: The abbreviation "Cbx. Ac." In the legend is not immediately obvious; I suggested writing out "carboxylic acids" on two lines instead for improved clarity. Response: Figure revised with "carboxylic acid" as legend entry. (revised page 10)

Please also note the supplement to this comment:
https://www.atmos-chem-phys-discuss.net/acp-2017-718/acp-2017-718-AC2-supplement.pdf

---

## Author Comment (AC3) · 16 Dec 2017

Temporally-Delineated Sources of Major Chemical Species in High Arctic Snow – Response to Anonymous Referee #3

Referee comments received and published: 23 October 2017 (quoted below)

We would like to thank Referee #3 for providing comments on this manuscript. We greatly appreciate the care with which the three referees have reviewed this manuscript and the improvements gained through their insight.

Response to Referee Discussion

Referee Comment: Review for Atom. Chem. Phys. Discuss. Temporally-Delineated

Source of Major Chemical Species in High Arctic Snow General review: The paper provides apportionment of chemical components in high Arctic snow, which is of interest. Some of the interpretation of source region and emission source connected to the PMF factors was not sufficiently supported and seemed stretched; this was particularly true for the discussion for the sulfate factor and the attribution of V, As and Se to dust/crustal materials in the dust factor. Improved consistency is needed for naming across the text, figures, and tables. I agree with comments provided by the previous referees. Response: We agree that factor naming should be consistent throughout. The revised manuscript uses the following names when referring to factors 1 to 7, respectively: sea salt, crustal metals, black carbon, carboxylic acids, nitrate, non-crustal metals, and sulphate. Please see the responses to referee #1 and #2 for specific replies to their comments.   Response to Detailed Comments

Referenced to Page/Line #(s) in the original manuscript:

3/5-7

Referee Comment: You need to give a bit more detail here, regardless of whether you are following previous protocol as this paper needs to be able to stand alone. How are these melted? How is the filtration accomplished? What is the storage protocol? How are the blanks?

Response: Additional details on the sample preparation and analysis have been provided in the revised supplemental, section S1.

Referee Comment: Please make this more explicit, especially for ones where the is temporal overlap in the peak concentration of the factor.

Original Line: The potential FLEXPART source regions associated with each PMF factor were identified.

Response: The calculation of the weighted FLEXPART source/influence regions is
described in Equation 4. (revised manuscript page/line(s): 6/4) In response to the existence of temporal overlap, we found that no two factors share more than two dates with peak above their respective 90% percentile. The text has been revised to note this (6/15-16). The highest correlation in factor contribution over time was seen between Factor 3 black carbon, Factor 5 nitrate, and Factor 6 non-crustal metals, with Pearson's correlation coefficient of 0.38 to 0.52. No other factors exhibited contribution correlation coefficients greater than 0.3. Furthermore, factors with similar peak periods may suggest similar source regions; thus the resultant similarities in the FLEXPART plots is not unexpected.

Referee Comment: Please make all factor names consistent: sea salt/marine sea salt/marine factor, choose one and use for all tables, text and figures.

Response: As per the response above, all references to the factors by name have been revised to be consistent.

Referee Comment: You should be able to find the ice extent for these specific time periods for the locations mentioned. Also, based on the heat map in Figure 3 for Factor 1 (you should really include the Factor names here as well, as it is difficult to keep track of which factor is which across a couple figures), the longest residential time is north of Greenland and Siberia – are these areas open water in January 2015? Wouldn't the open water have to have been close to the site for the correlation to local wind speed be relevant for sea spray sourcing?

Response: Per the referee's suggestion, sea ice concentration plots have been obtained from the NOAA G02135 archives (ftp://sidads.colorado.edu/DATASETS/NOAA/G02135/). Comparison of these plots and the potential source regions identified for Factor 1, sea salt, showed several

potential sources for sea salt: Barents Sea, Greenland Sea, Norwegian Sea, northern Atlantic, and portions of Baffin Bay and waters surrounding the Queen Elizabeth Islands. This information has been added to the manuscript (12/9-12).

Factor names have been added to Figure 3 (revised page 10).

The correlation between Factor 1, sea salt, and local wind speeds was weak, a Pearson's correlation of 0.28. We agree that for local wind speeds to be relevant there must be a local source of sea salt. This could include any local open water, blowing saline snow, or frost flowers; however, we would require more data to confirm the existence of any of these sources at the specified time. Upon further consideration, we have noted that Factor 1, sea salt, in fact has a stronger correlation with collection period length (Pearson's correlation coefficient of 0.47). The January peak of this factor was one of the longer collection period of the campaign. This may suggest that the deposition of sea salt aerosol was relatively continuous over time; thus longer collection periods were associated with higher sea salt signatures. However, it should be noted that both of these correlations are fairly weak (the 0.28 correlation has been deemed too weak to include in the revised manuscript), so these inferences should be considered uncertain. Section 3.2.1 of the manuscript has been revised to reflect the discussion above.

Figure 1

Referee Comment: Clarify whether these are soluble, insoluble or total metals.

Response: As stated in Section 3.1, only the portions of the ICP-MS metals considered insoluble were included in the PMF analysis. The caption for Figure 1 has been revised to restate this information.

Referee Comment: Make all factor names consistent throughout the manuscript: crustal metals vs dust. Also, the high contribution of V, As and Se might indicate anthropogenic pollution (i.e. coal or heavy oil combustion) not just "dust".

[Figure]

Response: Per comment above, factor names have been revised to be consistent throughout the text.

We agree that V, As, and Se are all typically thought of as anthropogenic in origin; yet, they all also exist in soils. This comment was addressed in response to Referee#1, copied below: While we agree that V, Se, and As are typical of anthropogenic sources they also occur in dust sources. The ratio of these metals to Al in Factor 2, crustal metals, were 0.0016, 0.0031, and 0.00081 m/m for V, Se, and As, respectively. Soils vary significantly in composition, but typical ratios to Al are 0.0012 - 0.0016, 0.000001 - 0.00027, and 0.00002 m/m for V, Se, and As, respectively (Taylor, 1964; Barrie, den Hartog, and Bottenheim, 1989; Masson-Delmotte et al., 2013). Measurements of local crustal sources in the Arctic have also seen ratios to Al of 0.0013 and 0.00013 m/m for V and As, respectively (Se not measured) (Barrie, den Hartog, and Bottenheim, 1989). As discussed in the manuscript, this gives enhancement ratios of approximately unity for V, 11-5000 for Se (note this large range is a result of the high variability in crustal measurements), and 6-37 for As. Thus, the loading of V in particular on this factor is very reasonable for a crustal source. The loadings of Se and As are higher than for typical soils but given the variability seen across crustal sources both could still be explained by a crustal source. Furthermore, the raw unapportioned concentration measurements of V, Se, and As all correlate to Al with Pearson's correlation coefficients of 0.91 or higher. Timeseries of these analytes are provided in the supplemental.

An important distinction in this analysis is that the V, Se, and As measurements being discussed are the insoluble portions (as noted in the original manuscript page 7 lines 31-32, and revised manuscript page 6 lines 2-3). The soluble portion of these metals was often below detection limits with weak signal-to-noise and therefore was excluded from the apportionment analysis (note that the portion considered as "soluble" would include soluble metals as well as insoluble metals associated with particles capable of passing through a 0.45 $\mu$m filter; Macdonald et al., 2017 provides further details about this analysis). Of these three metals soluble As had the highest number of measurements about detection limit. The soluble As time series correlated best with Factors 3 and 6, black carbon and non-crustal metals. The limited data available for soluble metals contributes a high degree of uncertainty to any discussion of their potential apportionment, but their correlation with these anthropogenic factors may indicate that the anthropogenic sources of these metals were mostly captured in the soluble measurements while the insoluble measurements represent a largely crustal source.

Figure 3

Referee Comment: The cyan diamonds and green triangles are very difficult to see.

Response: We agree that the symbols denoting Alert and volcanic sources are quite small (as noted by both referee #1 and 3). This was done so as to not block a significant portion of the trajectory plot. This figure will be uploaded as a high-resolution image allowing readers with difficulty seeing these symbols to simply zoom in as needed, without sacrificing the details of the trajectory plot.

Referee Comment: For Russian BC sources, there have been two new studies in the last year that should be included here and incorporated into the discussion:

Evans, Meredydd, Nazar Kholod, Teresa Kuklinski, Artur Denysenko, Steven J. Smith, Aaron Staniszewski, Wei Min Hao, Liang Liu, and Tami C. Bond. "Black carbon emissions in Russia: A critical review." Atmospheric Environment (2017).

Winiger, Patrik, August Andersson, Sabine Eckhardt, Andreas Stohl, Igor P. Semiletov, Oleg V. Dudarev, Alexander Charkin et al. "Siberian Arctic black carbon sources constrained by model and observation." Proceedings of the National Academy of Sciences (2017): 201613401.

Response: We thank the referee for the suggested references.

Evans et al. (2017) reviews a body of work related to BC sources within Russia. This
study develops a comprehensive budget of Russian BC emissions. Specifically, flaring and transportation are noted as major sources. Reference to the work by Evans et al. (2017) has been added to the manuscript. (14/33)

Winiger et al. (2017) is a study of the sources of BC to the Siberian Arctic from based on aerosol and isotope observations at Tiksi and comparison with dispersion modelling results. This paper highlighted the Autonomous Okrugs of Khanty-Mansi and Yamalo-Nenets regions as a hotspot for BC emissions, particularly in the winter months. This aligns with the regions of Russia noted as potential sources to Factor 3, black carbon, as shown in Figure 3 of the manuscript. Winiger et al also identified domestic and transportation activities as the major sources of BC to the Siberian Arctic (35% and 38%, respectively), with lower contributions from flaring, power plants, and open fires (6%, 9%, and 12%, respectively). Reference to the work by Winiger et al. (2017) has been added to the manuscript. (15/1)

Referee Comment: For detailed comparison with previous high Arctic snow apportionment studies, do also take into account more of the potential impact of Arctic location. The Hegg studies were quite different in the study design, representing PMF of a large number of Arctic sites as opposed to PMF at a single Arctic site.

Response: The difference in these studies is noted in Table 2 in the original document which lists the current study as temporally-refined and the Hegg studies as spatially-refined. However, per the suggestion of the referees, Table 2 has been removed from the revised text. The significance of location to BC source make-up has been noted in the revised text. (12/9-12 and 12/23-25)

Table 2

Referee Comment: Include location of the studies. The location is very relevant in terms of understanding BB impact across the Arctic. For the apportionment/co-

study develops a comprehensive budget of Russian BC emissions. Specifically, flaring and transportation are noted as major sources. Reference to the work by Evans et al. (2017) has been added to the manuscript. (14/33)

Winiger et al. (2017) is a study of the sources of BC to the Siberian Arctic from based on aerosol and isotope observations at Tiksi and comparison with dispersion modelling results. This paper highlighted the Autonomous Okrugs of Khanty-Mansi and Yamalo-Nenets regions as a hotspot for BC emissions, particularly in the winter months. This aligns with the regions of Russia noted as potential sources to Factor 3, black carbon, as shown in Figure 3 of the manuscript. Winiger et al also identified domestic and transportation activities as the major sources of BC to the Siberian Arctic (35% and 38%, respectively), with lower contributions from flaring, power plants, and open fires (6%, 9%, and 12%, respectively). Reference to the work by Winiger et al. (2017) has been added to the manuscript. (15/1)

Referee Comment: For detailed comparison with previous high Arctic snow apportionment studies, do also take into account more of the potential impact of Arctic location. The Hegg studies were quite different in the study design, representing PMF of a large number of Arctic sites as opposed to PMF at a single Arctic site.

Response: The difference in these studies is noted in Table 2 in the original document which lists the current study as temporally-refined and the Hegg studies as spatially-refined. However, per the suggestion of the referees, Table 2 has been removed from the revised text. The significance of location to BC source make-up has been noted in the revised text. (12/9-12 and 12/23-25)

Table 2

Referee Comment: Include location of the studies. The location is very relevant in terms of understanding BB impact across the Arctic. For the apportionment/co-

study develops a comprehensive budget of Russian BC emissions. Specifically, flaring and transportation are noted as major sources. Reference to the work by Evans et al. (2017) has been added to the manuscript. (14/33)

Winiger et al. (2017) is a study of the sources of BC to the Siberian Arctic from based on aerosol and isotope observations at Tiksi and comparison with dispersion modelling results. This paper highlighted the Autonomous Okrugs of Khanty-Mansi and Yamalo-Nenets regions as a hotspot for BC emissions, particularly in the winter months. This aligns with the regions of Russia noted as potential sources to Factor 3, black carbon, as shown in Figure 3 of the manuscript. Winiger et al also identified domestic and transportation activities as the major sources of BC to the Siberian Arctic (35% and 38%, respectively), with lower contributions from flaring, power plants, and open fires (6%, 9%, and 12%, respectively). Reference to the work by Winiger et al. (2017) has been added to the manuscript. (15/1)

Referee Comment: For detailed comparison with previous high Arctic snow apportionment studies, do also take into account more of the potential impact of Arctic location. The Hegg studies were quite different in the study design, representing PMF of a large number of Arctic sites as opposed to PMF at a single Arctic site.

Response: The difference in these studies is noted in Table 2 in the original document which lists the current study as temporally-refined and the Hegg studies as spatially-refined. However, per the suggestion of the referees, Table 2 has been removed from the revised text. The significance of location to BC source make-up has been noted in the revised text. (12/9-12 and 12/23-25)

Table 2

Referee Comment: Include location of the studies. The location is very relevant in terms of understanding BB impact across the Arctic. For the apportionment/co-

variance (again, use the same terms in the text and tables to avoid confusion), include types of species used in the modeling for BC apportionment.

Response: This table, Table 2 in the original manuscript, has been removed from the revised manuscript per the referees' suggestion.

Referee Comment: I think this sentence has been truncated "...linked with both biomass burning plumes..." and?

Original Line: Carboxylic acids within the Arctic have previously been linked with both biomass burning plumes (e.g., Jaffrezo et al., 1998; Legrand and de Angelis, 1996). Response: As also noted by referee #1 the word "both" in this sentence was a mistake. The sentence has been corrected to remove the word "both". (15/20-21)

Referee Comment: Where are source areas shown in Figure 2?

Response: The referee is correct that this line mistakenly referenced Figure 2 instead of Figure 3. This has been corrected in the revised manuscript. (15/3)

17/15-16

Referee Comment: It's not clear how this factor coincides with increased transport over the ice-free Norwegian Sea and northern Atlantic. Remove unless you can support.

Response: This section has been revised as follows to provide greater clarity: "Weighting the FLEXPART predicted source areas by the Factor 1 peak dates (Figure 3) showed the Eurasian coast of the Arctic Ocean, the Norwegian Sea, the Greenland Sea, and the northern Atlantic Ocean to be potential sources of sea salt to Alert. Ice-free areas were identified using the NOAA G02135 ice concentration images (retrieved from ftp://sidads.colorado.edu/DATASETS/NOAA/G02135/ November 2017). During periods of peak Factor 1, sea salt, contribution, the East Siberian Sea, Laptev Sea

and Kara Sea appear to have been largely ice-covered; however, the Barents Sea, Greenland Sea, Norwegian Sea, northern Atlantic, and portions of Baffin Bay and waters surrounding the Queen Elizabeth Islands all seem to have been ice-free or with new, thin ice coverage. Thus, sea salt spray from these areas likely contributed to the sea salt signal at Alert.". (12/9-16)

18/20-21

Referee Comment: The Flexpart in Figure 3 does not seem to match with the assignation of sulfate to volcanoes and the Smoking Hills.

Response: The Factor 7, Sulphate, section has been revised to address several comments from all referees. We recognize that Figure 3 does not show high influence from the noted volcanic sources for Factor 7; however, this plot only represents a ten-day back trajectory and does seem to indicate that Factor 7 is more likely a dominated by relatively local sources rather than long-range anthropogenic sources. Furthermore, these plots only highlight areas over which the trajectories passed within 500 m of the surface (as noted in section 2.4.2). This approach is useful for identifying ground-level sources which could have reasonably impacted the air mass. However, volcanic sources can impact air masses to a much great height, given the heat and velocity of the emitted plume; thus, trajectories at a greater height should be considered. We have reviewed the FLEXPART influence plot for Factor 7 for trajectories within 10 km of the surface and this plot does show greater potential influence from the BárÃřarbunga volcano in Iceland and the Smoking Hills in Canada. (section 3.2.7)

Page 18-19

Referee Comment: The explanation for the sulfate factor was a bit forced to match volcanism. If the metals factor was combined with sulfate in the six factor solution, it would seem that would indicate an anthropogenic source. When comparing to the connected Macdonald paper, the co-variance of sulfate and MSA (or MS, as it was called in the previous paper), might be spurious as MSA is only high in the early part

of the campaign.

Response: The six-factor solution produced a factor which roughly combined Factors 6 and 7 of the seven-factor solution; however, it does not reflect the observed seasonal trend in sulfate. The distinct fall peak in sulfate observed in this study is not predicted by the six-factor solution and as a result the sulfate predicted/measured fit is very poor (Pearson's correlation coefficient of only 0.38). The addition of the seventh factor enabled better recreation of the observed sulfate signal. The revised manuscript has been updated to include mention of this in the manuscript (19/7-10) and supplemental (Section S3.2 ad Figure S7).

Figure 4

Referee Comment: use the same naming for factors across all figures, text and tables. The abbreviation is difficult here.

Response: Figure 4 has been revised to use the full names for each factor.

Referee Comment: again, take location into account for comparison with other Arctic BC studies.

Response: Per the suggestion of the referees, the conclusions have been revised to reduce the focus on BC. The discussion of BC results has been changed to stress the importance of spatial and temporal variation in the BC sources.

Please also note the supplement to this comment:
https://www.atmos-chem-phys-discuss.net/acp-2017-718/acp-2017-718-AC3-supplement.pdf

---

## Author Response (AR2)

**Temporally-Delineated Sources of Major Chemical Species in High Arctic Snow – Second Response to Anonymous Referee #1**

Referee comments received and published: 4 January 2018 (quoted below in blue text)

**Response to Referee Comments – Main Text**

Referenced to Page/Line #(s) in the original manuscript:

**1/5**

Referee Comment: "this rich data provided" should be either: this rich data set, or these rich data

Response: Editorial comment revised as suggested. (revised manuscript page/line(s): 1/15)

**5/13**

Referee Comment: "limited number of snow samples measurements" should be either: snow sample measurements, or snow samples measured

Response: Editorial comment revised as suggested. (revised manuscript page/line(s): 5/13)

**Table 1**

Referee Comment: (and many times after) MSA is typically used to refer to methysulfonic acid. MS^- is used to refer to methylsulfonate (which is what is measured by IC in melted snow)

Response: The acronym MSA has been replaced with MS throughout the revised text, figures, and tables.

**Figure 1**

Referee Comment: Still unclear to me what the grey bars signify in this plot. If this is mass of analyte attributed to the factor, is it correct that calculating the ratio of the bar in one factor (for given analyte) over the sum of the 7 bars for that analyte would be the same as percentage shown with diamonds (on the other axis)? If yes, the bars add no information.

Response: The referee is correct in their interpretation of the grey bars in Figure 1: they represent the mass of each analyte loaded onto each factor and the individual mass associated to one factor divided by the total mass of that analyte is equivalent to the percentile loading presented as a black diamond on the same plot. This is simply two means of describing the same information. Given that the total mass loading differs across analytes, the percentile loading is useful in highlighting which factor dominates each analyte while the mass loading is useful in highlighting which analyte dominates each factor by mass. This information is repetitive but facilitates interpretation. This presentation is typical of many PMF papers.

Referee Comment: use of "enrichment ratios" thoughout section 3.2 is a little confusing. In this instance Cl/Na and K/Na have some precedent in the literature where enrichment factors defined as X/ssRef or X/crustalRef are used to assess whether X seems dominated by natural or anthropogenic sources. In this sense, comparing As/Al in snow to the ratio in "global typical" make sense as enrichment ratio, but the comparison to As/Al in local soil not so much. Likewise SO4/BC and NO3/BC are just ratios, since there is not really an accepted global mean composition of BB smoke (depends on way too many factor like fire stage, fuel type, moisture, etc.)

Response: The text has been revised to use the term "mass ratios" to describe the ratio of two analytes loaded onto the same factor. The term "enrichment ratio" is only used when comparing that calculated ratio to those observed by previous studies. (revised manuscript sections: 3.2.1 - 4)

**19/20**

Referee Comment: "within the northern Eurasia"---> within northern Eurasia

Response: Editorial comment revised as suggested. (revised manuscript page/line(s): 19/20)

**22/1-3**

Referee Comment: some kind of cut and paste error here, seems words are missing

Response: We thank the referee for noticing this error; this was a copy and paste mistake. The line has been revised as follows: "*Both V and Se are typically considered to be tracers of anthropogenic activity, specifically oil and coal combustion (Key and Hoggan, 1953; Rahn, 1971; Berg, Røyset, and Steinnes, 1994; Laing et al., 2014).*" (revised manuscript page/line(s): 22/1)

**Response to Referee Comments – Supplemental**

**S1.1**

Referee Comment: camo activities--->camp activites

were used to dividing the--->were used to divide

Response: Editorial comment revised as suggested. (revised supplemental section S1.1)

**S3.2**

Referee Comment: (under Fig S5)

with the exception of that--->with the exception that

As discussed in Chapter 5--->As discussed in 3.2.7 in main text

Response: Editorial comment revised as suggested. (revised supplemental section S3.2)